# Circulating exosomes suppress the induction of regulatory T cells via *let-7i* in multiple sclerosis

Kimitoshi Kimura[1,2], Hirohiko Hohjoh[3], Masashi Fukuoka[3], Wakiro Sato[1,4], Shinji Oki[1], Chiharu Tomi[1], Hiromi Yamaguchi[1], Takayuki Kondo[2,5], Ryosuke Takahashi[2] & Takashi Yamamura[1,4]

Multiple sclerosis (MS) is a T cell-mediated autoimmune disease of the central nervous system. Foxp3[+] regulatory T (Treg) cells are reduced in frequency and dysfunctional in patients with MS, but the underlying mechanisms of this deficiency are unclear. Here, we show that induction of human IFN-γ[−]IL-17A[−]Foxp3[+]CD4[+] T cells is inhibited in the presence of circulating exosomes from patients with MS. The exosomal miRNA profile of patients with MS differs from that of healthy controls, and *let-7i*, which is markedly increased in patients with MS, suppresses induction of Treg cells by targeting insulin like growth factor 1 receptor (*IGF1R*) and transforming growth factor beta receptor 1 (*TGFBR1*). Consistently, the expression of IGF1R and TGFBR1 on circulating naive CD4[+] T cells is reduced in patients with MS. Thus, our study shows that exosomal *let-7i* regulates MS pathogenesis by blocking the IGF1R/TGFBR1 pathway.

[1] Department of Immunology, National Institute of Neuroscience, National Center of Neurology and Psychiatry, 4-1-1 Ogawahigashi, Kodaira, Tokyo 187-8502, Japan. [2] Department of Neurology, Kyoto University Graduate School of Medicine, Yoshida-konoe-cho, Sakyo, Kyoto 606-8501, Japan. [3] Department of Molecular Pharmacology, National Institute of Neuroscience, National Center of Neurology and Psychiatry, 4-1-1 Ogawahigashi, Kodaira, Tokyo 187-8502, Japan. [4] Multiple Sclerosis Center, National Center of Neurology and Psychiatry, 4-1-1 Ogawahigashi, Kodaira, Tokyo 187-8551, Japan. [5] Department of Neurology, Kansai Medical University Medical Center, 10-15 Fumizono, Moriguchi, Osaka 570-8507, Japan. Correspondence and requests for materials should be addressed to T.Y. (email: yamamura@ncnp.go.jp)

Multiple sclerosis (MS) is an autoimmune disease of the central nervous system, in which inflammatory CD4[+] T helper 1 (Th1) and Th17 cells are pathogenic[1,2]. In mice, Foxp3[+] regulatory T (Treg) cells, originally defined as CD4[+]CD25[+] T cells, can suppress the proliferation and function of inflammatory T cells, thereby preventing the development of autoimmune diseases[3]. Disruption of the balance between inflammatory and regulatory T cells may thus underlie MS pathogenesis[4]. Previous studies have found a reduced frequency of CD4[+]CD25[+]CD127[−]Foxp3[+] Treg cells in the peripheral blood of patients with MS[5,6], as well as functional impairment of CD4[+]CD25[+]Foxp3[+] Treg cells in patients with MS[7,8]. However, the underlying mechanisms of alterations of Treg cells are unclear.

Micro RNAs (miRNA) are small non-coding RNAs that function as post-transcriptional regulators of gene expression by inhibiting translation of messenger RNAs (mRNA). Previous studies have revealed critical functions of miRNAs in the differentiation and function of helper T cells[9,10]. Alterations of miRNAs in the circulation, inflammatory cell populations or pathological samples of autoimmune diseases have also been documented[11,12]. MiRNAs can be present in exosomes, which are extracellular vesicles (EV) smaller than 150 nm in diameter. Exosomes can affect the target cells via gene regulation, which is mediated by transfer of miRNAs[13–15]. Exosomes are secreted from various cell types into circulation, and are delivered to target cells throughout the body. Gene regulation with exosomes, in which extrinsic miRNAs exert direct effect on target genes in recipient cells, is regarded as a form of intercellular communication, which differs from conventional communication by cytokines and cell surface molecules[15]. Critical involvement of exosomes has been demonstrated in various human disorders, including cancer and neurodegenerative diseases[16,17]. Approximately 100 miRNAs have been shown to be dysregulated across various tissues, including brain, blood and cerebrospinal fluid from patients with MS, but pathological effects have only been reported for a small subset of these miRNAs[12,18–20]. Additionally, exosomal miRNA function has not been studied in MS.

In this study, we isolate circulating exosomes from the blood of patients with MS and evaluate potential pathogenic function of these miRNA-containing exosomes in MS. We find that exosomes derived from patients with MS (MS-exosome) can selectively affect IFN-γ[−]IL-17A[−]Foxp3[+]CD4[+] Treg cells in vitro. Several miRNAs are more abundant in the MS-exosome than in exosomes from healthy donors. Among those upregulated in patients with MS, let-7i can suppress Treg cell induction by inhibiting the expression of insulin like growth factor 1 receptor (IGF1R) and transforming growth factor beta receptor 1 (TGFBR1). Our findings imply that altered miRNA expression in MS-exosome may contribute to the pathogenesis by disrupting the homeostasis of Treg cells.

## Results

**Treg cell frequency is decreased by MS-exosome.** Exosomes are secreted from various cell types, circulate in the body, and alter the function of the recipient cells via delivery of the exosomal miRNAs. To investigate the function of the circulating exosomes carrying miRNAs in MS, we purified exosomes from the plasma of healthy controls (HC) and patients with MS. The average size of the vesicles purified from the plasma samples was 96.5 nm ($n = 6$, standard deviation: 17.9 nm). Most of the collected vesicles were typical in size as exosomes, which are smaller than 150 nm (Fig. 1a)[15]. Furthermore, they expressed conventional exosome markers, CD9 and CD63, but lacked the expression of Cytochrome c, a mitochondrial protein, which is not present in exosomes (Fig. 1b)[21].

Given the critical contribution of cellular immunity in MS, here, we focused on the possible action of exosomes against T cells. We cultured peripheral blood CD3[+] T cells from HC under stimulation with anti-CD3 and anti-CD28 monoclonal antibodies (mAb) in the presence of exosomes derived from patients with MS (MS-exosome) or those from HC (HC-exosome). Notably, after culture with MS-exosome, the frequency of Foxp3[+]CD4[+] T cells was significantly reduced as compared to that after culture with HC-exosome ($p < 0.01$, a one-way analysis of variance (ANOVA)) (Fig. 1d). When the Foxp3[+]CD4[+] T cells were phenotypically subdivided, the frequency of IFN-γ[−]IL-17A[−]Foxp3[+]CD4[+] T cells, regarded as most suppressive Treg cells[8,22,23], reduced ($p < 0.05$, a one-way ANOVA), whereas the frequency of Foxp3[+]CD4[+] T cells secreting IFN-γ or IL-17A did not differ significantly between the two groups (Fig. 1c, d). It is assumed that the majority of Foxp3[+]CD4[+] T cells secreting the inflammatory cytokines are activated inflammatory T cells in which Foxp3 expression is transient, or those representing dysfunctional Treg cells with impaired suppressive function which are induced in the inflammatory cytokine milieu[8,22–25]. To provide further confirmation, we analysed the DNA methylation status of the signal transducer and activator of transcription 5 (STAT5)-responsive region of the *FOXP3* gene in these T cell populations. CpG sites in the region are known to be demethylated in functional Foxp3[+] Treg cells[23]. The region examined was moderately demethylated in IFN-γ[−]IL-17A[−]Foxp3[+]CD4[+] T cells, and clearly different from that of Foxp3[+]CD4[+] T cells secreting IFN-γ or IL-17A, which was methylated to an extent similar to that of Foxp3[−] non-Treg cells (Supplementary Fig. 1a, b). Since IFN-γ[−]IL-17A[−]Foxp3[+]CD4[+] T cells cannot be acquired intact, we could not assess their regulatory function directly. Instead, a pool of CD25[+]CD127[−]CD49d[−]CD4[+] T cells enriched in IFN-γ[−]IL-17A[−]Foxp3[+]CD4[+] T cells were isolated (Supplementary Fig. 1c). Subsequently, this population was shown to possess a regulatory function against responder cells (CD45RA[+]CD25[−]CD4[+] T cells) in vitro (Supplementary Fig. 1d). These data indicated that the IFN-γ[−]IL-17A[−] cells are the functional Treg cell population among all Foxp3[+]CD4[+] T cells, whereas the remaining IFN-γ[+] or IL-17A[+] cells are not. This is consistent with previous studies[8,22–25]. There was no significant difference in the frequency of potentially inflammatory CD4[+] T cells such as those secreting IFN-γ, IL-17A or GM-CSF, after culture with HC-exosome or MS-exosome (Fig. 1d; Supplementary Fig. 2). The frequency of IL-10[+]CD4[+] T cells, which have regulatory activity, was not altered by the presence of MS-exosome (Supplementary Fig. 2).

Next, we cultured CD3[+] T cells in the presence or absence of HC-exosome labelled with PKH67 green fluorescence lipid dye. After 24 h, PKH67 fluorescence was detected on the T cells cultured with the labelled exosomes, but not on those cultured without exosomes, supporting the occurrence of interaction between exosomes and the T cells (Supplementary Fig. 3a, b).

**Let-7i is increased in the exosomes from patients with MS.** To understand how MS-exosome affects Treg cells, we further analysed miRNA expression profiles in MS-exosome as compared to those in HC-exosome. Results of microarray analysis, demonstrated by cluster analysis and principal component (PC) analysis, showed remarkable differences in miRNA profiles between MS-exosome and HC-exosome (Fig. 2a, c), indicating the potential value of the exosomal miRNA analysis for the diagnosis of MS. Subsequently, we selected four miRNAs, let-7i, miR-19b, miR-25 and miR-92a for further analysis, the expression of which showed most significant differences between MS-exosome and HC-exosome (Fig. 2b). The upregulation of these miRNAs was

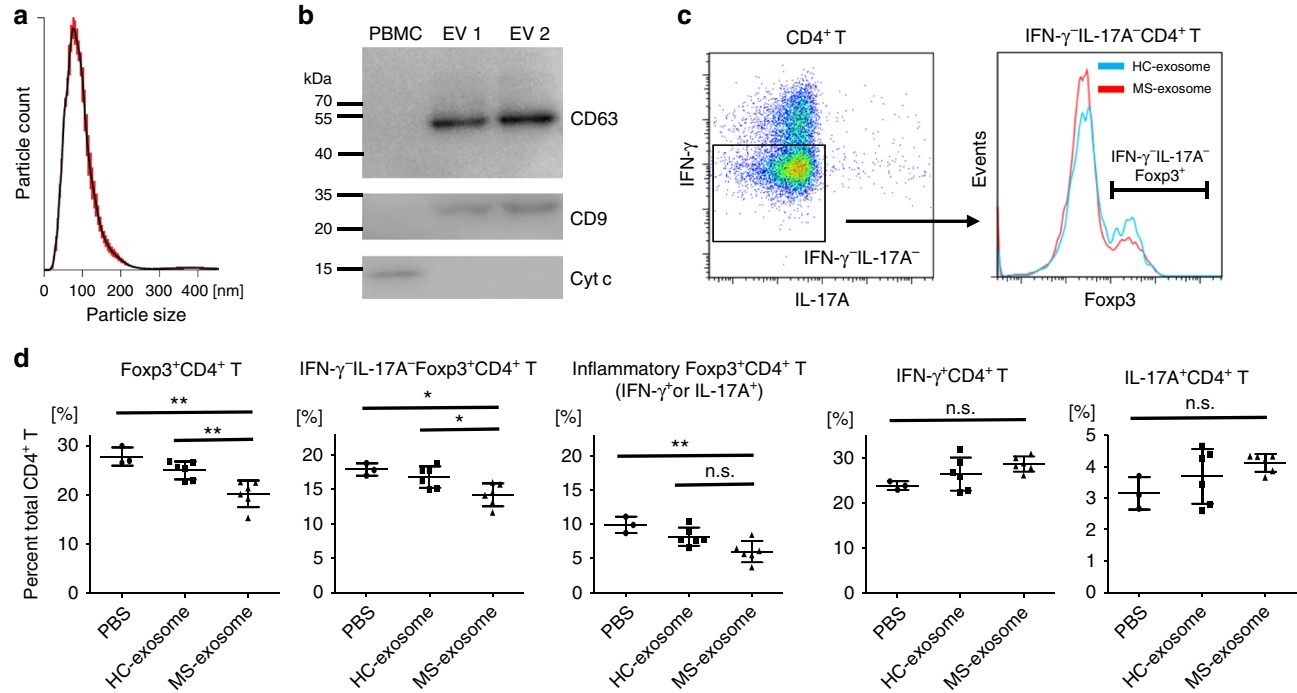

**Fig. 1** Exosomes from patients with MS can selectively decrease the frequency of Treg cells **a** Representative size distribution of purified exosomes. Using a NanoSight LM10 nanoparticle analysis system, the size was analysed three times for each sample. Red error bars indicate the standard error of the mean. **b** Western blot analysis for CD9, CD63 and Cytochrome c proteins. The PBMC and exosome samples were collected from HC. Each lane was loaded with 3 μg of protein for blotting. **c** Dot plot and histogram of flow cytometry data. T cells were prepared from the peripheral blood of a healthy volunteer. They were cultured with PBS as a control or with exosomes derived from HC (HC-exosome) or patients with MS (MS-exosome) under stimulation with anti-CD3 and anti-CD28 mAbs for 48 h. IFN-γ⁻IL-17A⁻CD4⁺ T cells were defined as shown in the left panel, and then the expression of Foxp3 was evaluated as shown in the right panel. **d** The frequencies of inflammatory and regulatory T cells among CD4⁺ T cells after the culture described above. Among Foxp3⁺CD4⁺ T cells, IFN-γ⁻IL-17A⁻Foxp3⁺CD4⁺ T cells are known to represent most effective Treg population[8, 22, 23], whereas Foxp3⁺CD4⁺ T cells secreting IFN-γ or IL-17A are supposed to be dysfunctional in the suppressive activity[8, 22–25]. Data are representative of two independent experiments. A one-way ANOVA with Bonferroni's comparison test was used for statistical analysis. Error bars represent the mean ± s.d. *$p < 0.05$, **$p < 0.01$. s.d. standard deviation, n.s. not significant, ANOVA analysis of variance, Cyt c cytochrome c, PBMC peripheral blood mononuclear cell, EV extracellular vesicle, PBS phosphate-buffered saline

confirmed by reverse transcription quantitative polymerase chain reaction (RT-qPCR) ($p = 0.0005$, 0.001, 0.0004 and 0.001, respectively, an unpaired $t$ test) (Fig. 2d). The total amount of RNAs in exosomes was not significantly different between HC and patients with MS (Fig. 2e). We expected that monitoring expression levels of these miRNAs might prove to be useful for distinguishing MS subtypes and evaluating the clinical activity. However, no significant differences were observed between relapsing-remitting MS (RRMS) and secondary-progressive MS (SPMS), or between MS in remission and relapse phases (Fig. 2d). Sex and age had no significant effects on the expression levels of the four miRNAs (Supplementary Fig. 4a, b). These results indicated that the exosomal miRNA profile characteristic for MS may be maintained after being shaped at a preclinical or very early stage of the disease.

**Treg cell frequency inversely correlates with exosomal *let-7i*.** We evaluated if the relationship might exist between the frequency of the IFN-γ⁻IL-17A⁻Foxp3⁺CD4⁺ Treg cells after culture (Fig. 1d) and the amount of miRNAs in the exosomes added to the culture (Fig. 2d). It was first noted that the amount of *let-7i* in each exosome sample negatively correlated with the frequency of IFN-γ⁻IL-17A⁻Foxp3⁺CD4⁺ Treg cells ($p = 0.012$, Pearson's correlation analysis) (Fig. 3a). This raised a possibility that *let-7i* may participate in the bias for reduction of the Treg cell frequency. A trend for negative correlation between *miR-19b* and Treg cells was also observed, but it was not statistically significant ($p =$

0.075, Pearson's correlation analysis). No trends for correlation were seen between the Treg cell frequency and either of the remaining miRNAs (Fig. 3a). Because there was no difference in the amount of total RNAs between HC and patients with MS (Fig. 3b), we suspected that *let-7i* overexpressed in MS-exosome might have a key function in the suppression of the IFN-γ⁻IL-17A⁻Foxp3⁺CD4⁺ Treg cells in vitro. Additionally, we examined the ratio of mature *let-7i* to primary *let-7i* in the T cells cultured with HC-exosome or MS-exosome. MiRNA genes are transcribed into primary miRNAs, which are then processed into mature miRNAs during several steps in the nucleus and cytoplasm[9]. Therefore, the mature *let-7i*/primary *let-7i* ratio in the cells could be higher, if they actually uptake exogenous mature *let-7i*. Higher mature *let-7i*/primary *let-7i* ratios were observed in T cells cultured with MS-exosome as compared to those cultured with HC-exosome ($p = 0.022$, an unpaired $t$ test) (Fig. 3c). This observation is consistent with the hypothesis that exosomal *let-7i* is actually taken up by T cells and may exert some effects.

**Treg cells but not Th1 or Th17 cells are affected by *let-7i*.** To evaluate the functionality of the miRNAs upregulated in MS-exosome, CD3⁺ T cells from healthy donors were transfected with *let-7i*, *miR-19b*, *miR-25* or *miR-92a*, and cultured with anti-CD3 and anti-CD28 mAbs for 72 h. Subsequently, the frequencies of inflammatory and regulatory T cell populations were evaluated by flow cytometer. Although the frequencies of IFN-γ⁺CD4⁺ T cells and IL-17A⁺CD4⁺ T cells were not influenced by transfection

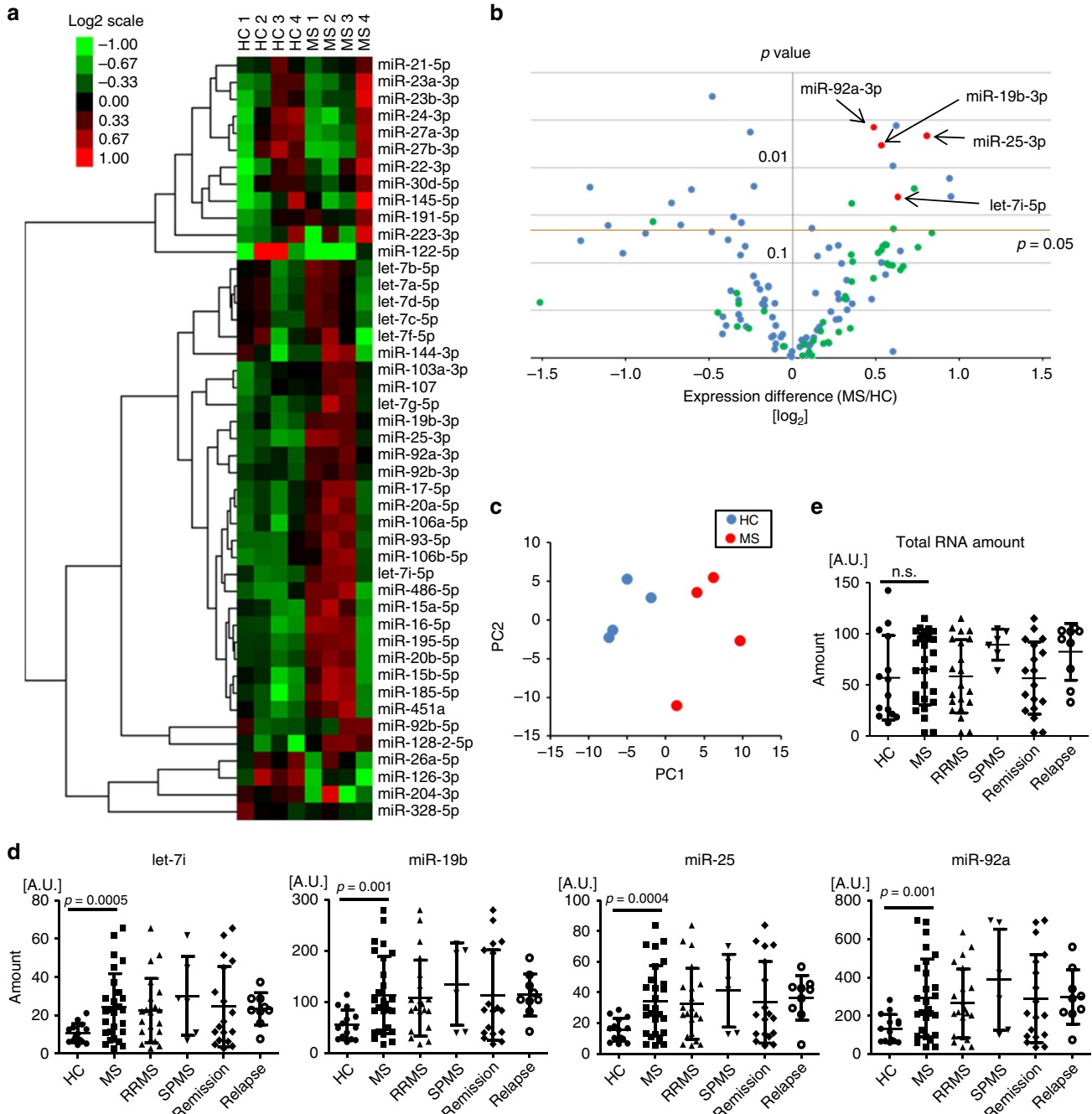

**Fig. 2** Exosomal miRNA profile differentiates patients with MS from healthy controls **a** Heat map of miRNA expression profile in the exosomes from HC and patients with MS. RNA was extracted from exosomes, which were isolated from the plasma of four HC and four patients with MS. A 3D-Gene Human miRNA Oligo chip (TORAY) was used for microarray analysis. MiRNAs with assigned identification number lower than 500 and signal intensity higher than 100 were selected and clustered based on the expression patterns of the eight samples. **b** Volcano plot of miRNAs in the exosomes. The expression difference of each miRNA between MS-exosome and HC-exosome is plotted on the X axis in log2 scale. p value of the difference by t test is plotted on the Y axis. Blue dots represent miRNAs with signal intensity higher than 100. Green dots represent miRNAs with identification number lower than 500 and signal intensity higher than 100. Four miRNAs pointed by arrows (red dots) were identified as candidate miRNAs upregulated in patients with MS. They were selected from green dots. **c** PC analysis for the MS and HC samples based on the miRNA expression profiles, using the same data as in **b**. **d,e** Quantification of the candidate miRNAs by RT-qPCR. The total amount of RNAs in the exosomes from the same amount of plasma was also examined. Based on the clinical information, patients with MS were divided into those with RRMS or SPMS and into those in remission or relapse phase. An unpaired t test was used for statistical analysis. Error bars represent the mean ± s.d. s.d. standard deviation, n.s. not significant, PC principal component, A.U. arbitrary unit, RT-qPCR reverse transcription quantitative polymerase chain reaction, RRMS relapsing-remitting MS, SPMS secondary-progressive MS

with the miRNAs, the frequency of IFN-γ⁻IL-17A⁻Foxp3⁺CD4⁺ Treg cells was significantly decreased after transfection with *let-7i* ($p < 0.05$, a one-way ANOVA) (Fig. 4a). In contrast, Foxp3⁺CD4⁺ T cells secreting IFN-γ or IL-17A were not affected. Next, to

examine the involvement of *let-7i* in the effect of exosomes on Treg cells, T cells were transfected with *let-7i* inhibitor and cultured with exosomes as in Fig. 1d. No difference was noted in the frequency of IFN-γ⁻IL-17A⁻Foxp3⁺CD4⁺ Treg cells between

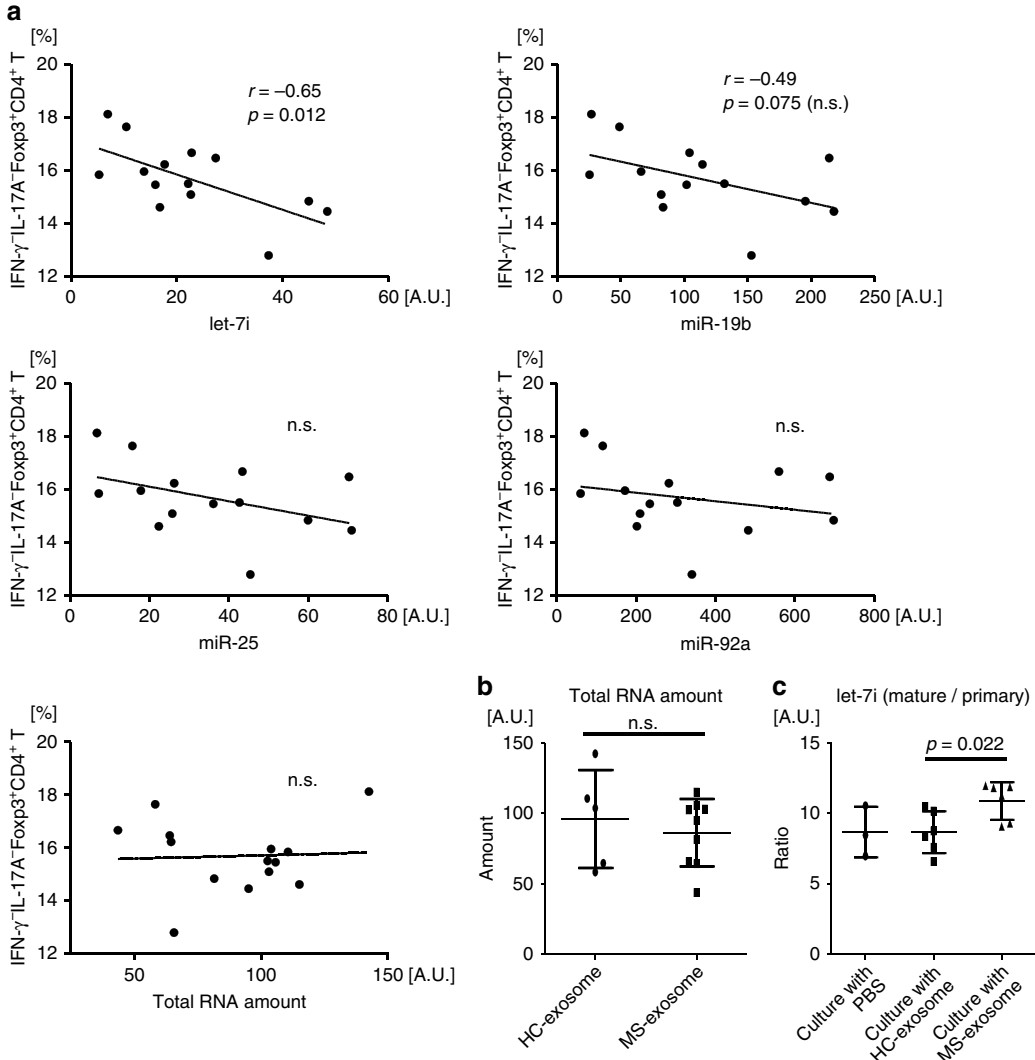

**Fig. 3** Expression of *let-7i* in exogenously added exosomes negatively correlates with Treg cell frequency. **a** Correlation between the frequency of IFN-γ⁻IL-17A⁻Foxp3⁺CD4⁺ Treg cells and exosomal miRNAs. The same experiment as that described in the legend for Fig. 1 was repeated with a different set of primary T cells and exosome samples. Along with the evaluation of the effect of exogenous exosome on Treg cells in the culture, we also quantified the expression levels of miRNA in each added exosome sample. Here, we analysed the relationship between the miRNA levels and the frequency of IFN-γ⁻IL-17A⁻Foxp3⁺CD4⁺ Treg cells. **b** The amount of total RNA. We also measured total RNA in the exosomes added to the cultures, showing no significant difference between HC and patients with MS. **c** Expression levels of mature *let-7i* and primary *let-7i* were evaluated by RT-qPCR in the T cells after culture with PBS, HC-exosome or MS-exosome. Pearson's correlation analysis was used in **a**, and an unpaired *t* test was used in **b** and **c** for statistical analysis. Error bars represent the mean ± s.d. s.d. standard deviation, n.s. not significant, A.U. arbitrary unit, RT-qPCR reverse transcription quantitative polymerase chain reaction, PBS phosphate-buffered saline

HC-exosome and MS-exosome groups after treatment with the *let-7i* inhibitor (Fig. 4b). The rate of increase in the Treg cell frequency after treatment with the *let-7i* inhibitor was greater in the presence of MS-exosome than HC-exosome ($p = 0.020$, an unpaired *t* test) (Fig. 4b), which was consistent with the result that *let-7i* was more abundant in MS-exosome (Fig. 2d). Given the inverse correlation between exosomal *let-7i* and Treg cell frequency (Fig. 3a), the present result indicated that Treg cells are the potential targets of *let-7i*, which is upregulated in MS-exosome.

**MS-exosome and *let-7i* inhibit Treg cell differentiation**. To further investigate the population that was influenced by MS-exosome and *let-7i* and is responsible for the reduced frequency of Treg cells after culture, four CD4⁺ T cell populations, including

naive, memory and Treg cells, were sorted based on the expression of CD25 and CD45RA. While naive and memory CD4⁺ T cells were sorted as CD45RA⁺CD25⁻ and CD45RA⁻CD25⁻CD4⁺ T cells, resting Treg and activated Treg cells were defined as CD45RA⁺CD25⁺ and CD45RA⁻CD25^high CD4⁺ T cells, respectively (Fig. 5a). These Treg cells have been previously shown to possess sufficient regulatory function[23]. Each isolated population was cultured with HC-exosome or MS-exosome under stimulation with anti-CD3 and anti-CD28 mAbs for 72 h. MS-exosome specifically inhibited the differentiation of IFN-γ⁻IL-17A⁻Foxp3⁺ Treg cells from naive CD4⁺ T cells as compared to HC-exosome ($p = 0.003$, an unpaired *t* test) (Fig. 5b). In contrast, there was no difference in the survival and proliferation of resting Treg and activated Treg cells (Fig. 5b). The frequency of IFN-γ⁻IL-17A ⁻Foxp3⁺ Treg cells seemed to decrease among memory CD4⁺ T cells cultured with MS-exosome, but this was not statistically

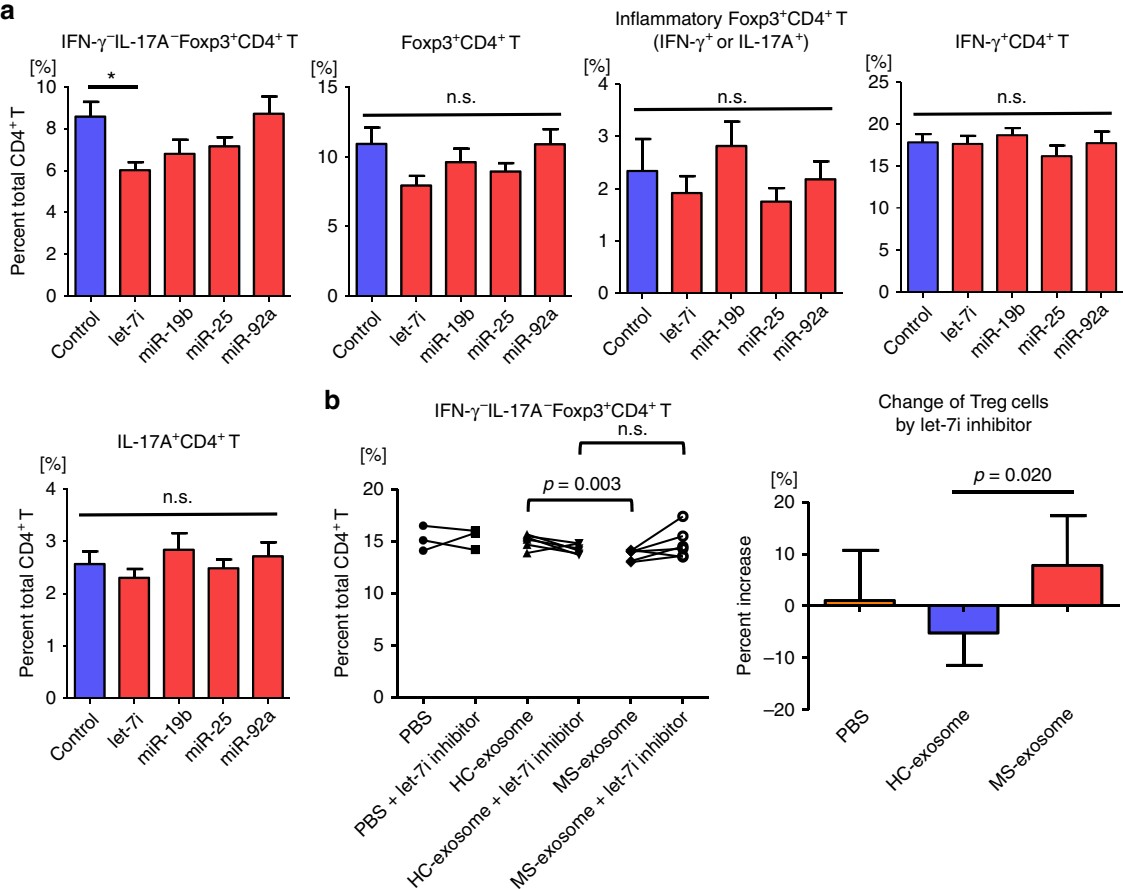

**Fig. 4** *Let-7i* transfection decreases the frequency of Treg cells. **a** T cells were transfected with *let-7i*, *miR-19b*, *miR-25* or *miR-92a* and then cultured under stimulation with anti-CD3 and anti-CD28 mAbs for 72 h. The phenotype was determined by flow cytometry based on cytokine expression (IFN-γ, IL-17A) and transcription factor expression (Foxp3). Data are representative of two independent experiments. **b** T cells were transfected with *let-7i* inhibitor and cultured in the presence of MS-exosome or HC-exosome under stimulation with anti-CD3 and anti-CD28 mAbs for 72 h. The frequencies of IFN-γ−IL-17A−Foxp3+CD4+ Treg cells were analysed and the rates of increase by *let-7i* inhibitor were plotted. Data are representative of two independent experiments. *n* = 6 in each group in **a**, *n* = 6 in each group with exosomes and *n* = 3 in the groups with PBS in **b**. A one-way ANOVA with Dunnett's comparison test was used in **a**, and an unpaired *t* test was used in **b** for statistical analysis. Error bars represent the mean ± s.e.m. in **a**, and the mean ± s.d. in **b**. *$p < 0.05$. s.e.m. standard error of the mean, s.d. standard deviation, n.s. not significant, ANOVA analysis of variance, PBS phosphate-buffered saline

significant (Fig. 5b). Consistent with these findings, *let-7i* inhibited the induction of IFN-γ−IL-17A−Foxp3+ Treg cells from naive CD4+ T cells ($p = 0.006$, an unpaired *t* test), and no significant change in Treg cells was noted among cultured memory CD4+ T cells, as well as in the survival and proliferation of resting and activated Treg cells (Fig. 5c). Next, mature and primary *let-7i* were quantified in CD45RA+ naive CD4+ T cells in the peripheral blood of HC and patients with MS to examine whether the amount is changed in vivo. The ratio of mature *let-7i* to primary *let-7i* increased in patients with MS ($p = 0.012$, an unpaired *t* test) (Fig. 5d), raising a possibility that exogenous *let-7i* was acquired by naive CD4+ T cells. Furthermore, the mature *let-7i*/primary *let-7i* ratio positively correlated with the amount of mature *let-7i* in exosomes from the blood ($p = 0.042$, Pearson's correlation analysis) (Fig. 5d). These results suggested that *let-7i* in circulating exosomes is actually taken up by naive CD4+ T cells and then would inhibit their differentiation into functional Treg cells in vivo.

**Let-7i suppresses the expression of IGF1R and TGFBR1.** To understand the mechanism of *let-7i*-mediated inhibition of Treg cell induction, we have attempted to identify target genes of *let-7i*. Using an online database for target prediction, TargetScanHuman (Release 6.2, 7.0 and 7.1)[26], we were able to identify candidate

genes targeted by *let-7i*. Subsequently, we selected suppressor of cytokine signalling 1 (*SOCS1*), *IGF1R* and *TGFBR1* for further analysis, as they are known to be associated with homeostasis of Treg cells. A previous study suggested that restriction of IFN-γ-STAT1 (signal transducer and activator of transcription 1) signalling by SOCS1 is critical for stabilising Foxp3+ Treg cells[27]. Accordingly, *Socs1*-deficient mice developed a T cell-mediated autoimmune inflammatory disease, as a result of defective Treg functions characterised by an increase in phosphorylated STAT1 (pSTAT1)[27,28]. However, neither pSTAT1 nor STAT1 in CD4+ T cells was changed after transfection of *let-7i* in our culture system (Supplementary Fig. 5), implying that SOCS1 does not have a major function in the *let-7i*-mediated suppression of Treg cells.

TGFβ has an important function during the differentiation of Treg cells[29], whereas IGF1 is known to expand Treg cell populations in vitro, thereby inhibiting autoimmune diseases and allergies[30,31]. Using blood samples from healthy volunteers, it was confirmed that exogenous IGF1 significantly promoted TGFβ-dependent differentiation of naive CD4+ T cells into IFN-γ−IL-17A−Foxp3+CD4+ Treg cells ($p = 0.031$, an unpaired *t* test) (Fig. 6a). Given that TGFβ- and IGF1-signalling pathways promote induction of Treg cells, we evaluated the effect of *let-7i* on IGF1R and TGFBR1 expressed by T cells. CD3+ T cells were

transfected with *let-7i* and cultured in the presence of anti-CD3 and anti-CD28 mAbs for 72 h. Notably, the expression of IGF1R and TGFBR1 on the cultured CD4[+] T cells was significantly decreased ($p = 0.005$ and 0.001, respectively, an unpaired *t* test) (Fig. 6b). This is consistent with the premise that the 3'

untranslated regions (3'UTR) of these genes are targets of let-7 family[32,33]. To examine the involvement of IGF1R and TGFBR1 in the differentiation of Treg cells from naive T cells, naive CD4[+] T cells were transfected with siRNAs targeting *IGF1R* and *TGFBR1*, and then cultured in the presence of TGFβ and IL-2 for

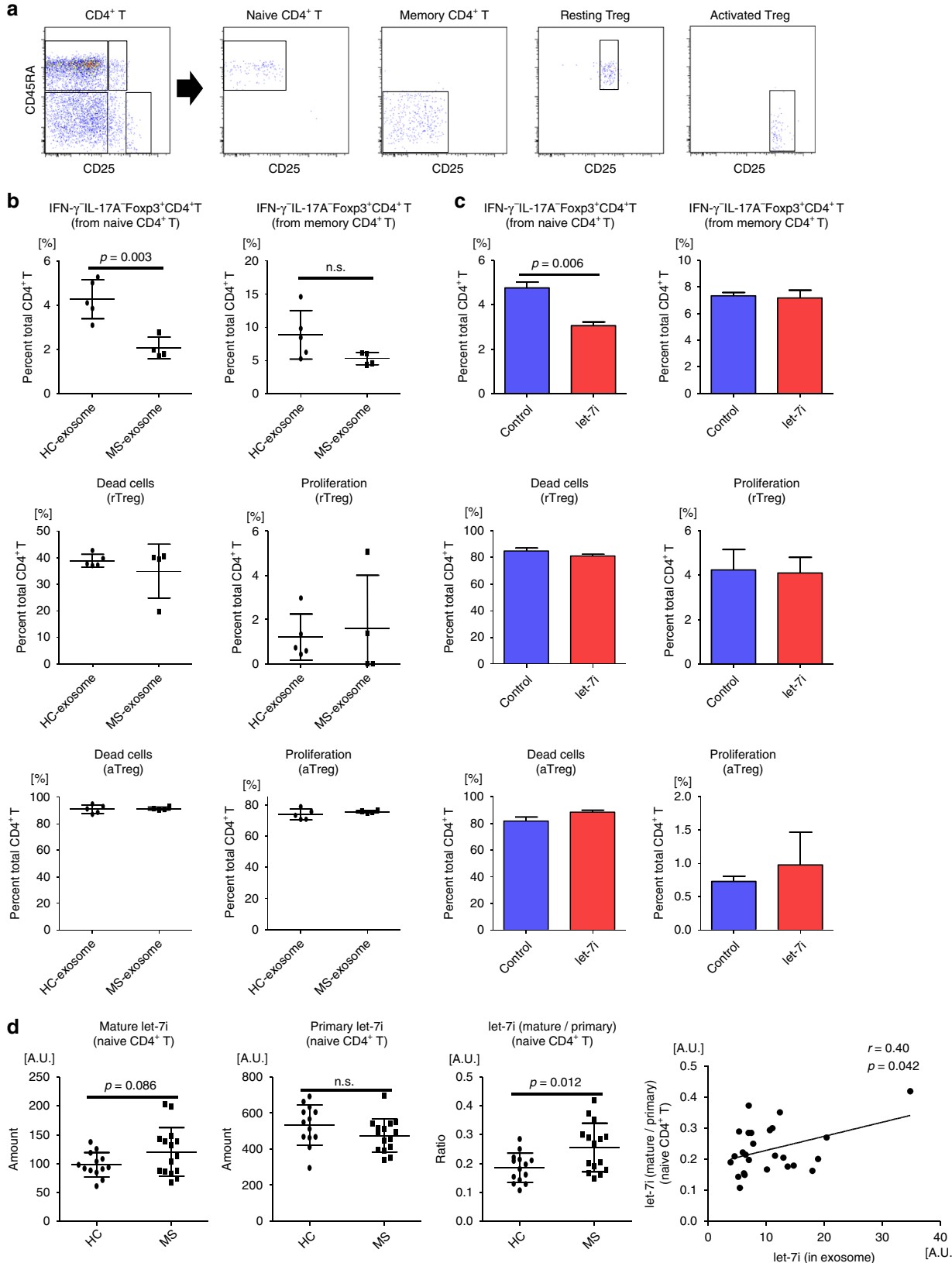

72 h. Knockdown of either or both of the receptors remarkably inhibited the differentiation of Treg cells ($p < 0.01$, $p < 0.001$ and $p < 0.001$, respectively, a one-way ANOVA) (Fig. 6c), indicating that the IGF1R and the TGFBR1 pathways collaboratively promote induction of Treg cells during their differentiation stage.

**Knockdown of *IGF1R* and *TGFBR1* decreases Treg cell frequency**. The in vitro analysis has not only confirmed the function of IGF1R and TGFBR1 in the differentiation of Treg cells, but also raised the possibility that *let-7i*-mediated inhibition of TGFβ signalling and IGF1 signalling might account for the reduction of the Treg cell frequency in patients with MS, because *let-7i* was also shown to inhibit differentiation of Treg cells (Fig. 5c). To evaluate the relevance of this speculation, peripheral blood CD3$^+$ T cells were transfected with various siRNAs and cultured for 72 h before flow cytometry analysis. It was confirmed that knockdown of *IGF1R* with three different siRNAs resulted in a similar reduction in the frequency of IFN-γ$^-$IL-17A$^-$Foxp3$^+$CD4$^+$ Treg cells ($p < 0.05$ or $p < 0.01$, a one-way ANOVA), whereas IFN-γ$^+$ or IL-17A$^+$ CD4$^+$ T cells were not reduced (Fig. 7a). Next, we transfected T cells with *TGFBR1*-specific siRNAs in the presence or absence of *IGF1R*-siRNA. The frequency of the Treg cells was reduced after knocking-down *TGFBR1*, although it was not statistically significant (Fig. 7b). We could neither reveal the additive effects of *TGFBR1* knockdown on *IGF1R* knockdown in reducing Treg cells, nor could we show the significant effects of *TGFBR1*-specific siRNAs alone or with *IGF1R*-specific siRNAs on inflammatory CD4$^+$ T cells. In addition, TGFβ neutralisation cancelled the reduced frequencies of Treg cells by knockdown of *TGFBR1* or *IGF1R* (Supplementary Fig. 6a and b), supporting a fundamental function of TGFβ in the differentiation of Treg cells as suggested in Fig. 6a.

**TGFBR1 and IGF1R expression on T cells is decreased in MS**. To obtain supportive evidences for the alteration of *let-7i-IGF1R/TGFBR1* axis in patients with MS in vivo, we isolated peripheral blood mononuclear cells (PBMC) from HC and patients with MS. Consistently with previous reports[5,6], the frequency of IFN-γ$^-$IL-17A$^-$Foxp3$^+$ Treg cells among memory CD4$^+$ T cells was significantly decreased in patients with MS as compared to that in HC ($p = 0.047$, an unpaired $t$ test) (Fig. 8a). Strikingly, it was found that the expression levels of TGFBR1 and IGF1R on CD45RA$^+$ naive CD4$^+$ T cells were significantly decreased in patients with MS ($p = 0.041$ and 0.012, respectively, an unpaired $t$ test) (Fig. 8b, c). When the samples were grouped based on the amount of *let-7i* in circulating exosomes (*let-7i* high and low; each contains the same number of samples), the expression of TGFBR1 on naive CD4$^+$ T cells was significantly lower in the *let-7i* high group than in the *let-7i* low group ($p = 0.025$, an unpaired $t$ test) (Fig. 8d). Significant negative correlation between them was also detected ($p = 0.020$, Pearson's correlation analysis) (Fig. 8d). Similarly, the persons with higher amounts of exosomal *let-7i* (*let-*

*7i* high) had decreased frequencies of IFN-γ$^-$IL-17A$^-$Foxp3$^+$CD4$^+$ Treg cells in circulation ($p = 0.043$, an unpaired $t$ test), although the negative correlation was not significant (Fig. 8d). This discrepancy might indicate the involvement of factors other than exosomal *let-7i* in the homeostasis of Treg cells, although it is possible that validation with a larger sample size might blur the discrepancy. Further analysis revealed a strong negative correlation between the amount of *let-7i* in naive CD4$^+$ T cells and IFN-γ$^-$IL-17A$^-$Foxp3$^+$CD4$^+$ Treg cells ($p = 0.0007$, Pearson's correlation analysis) (Fig. 8e). In addition, a negative correlation was observed between the expression of IGF1R and the amount of *let-7i* in naive CD4$^+$ T cells ($p = 0.011$, Pearson's correlation analysis). There was also a trend for inverse correlation between the expression of TGFBR1 and the amount of *let-7i* in naive CD4$^+$ T cells, though statistically insignificant ($p = 0.081$, Pearson's correlation analysis) (Fig. 8e). Moreover, a significant correlation was detected between the Treg cell frequency and the expression of TGFBR1 or IGF1R on naive CD4$^+$ T cells ($p = 0.043$ and 0.002, respectively, Pearson's correlation analysis) (Fig. 8f). These ex vivo data are consistent with the in vitro studies, which showed that *let-7i*-induced downregulation of IGF1R and TGFBR1 (Fig. 6b) may account for the inhibition of Treg cell induction from naive CD4$^+$ T cells (Fig. 6c). Furthermore, this level of consistency between ex vivo and in vitro results would allow us to hypothesise that *let-7i* upregulated in MS-exosome may act on naive CD4$^+$ T cells, and thereby cause the phenotypic changes in those cells, resulting in the reduction of induced Treg cell frequency in vivo. Relevance of this model should be tested in the future by using various means, including clinical interventions targeting exosomes and miRNAs.

## Discussion

The implications of miRNAs in the pathogenesis of intractable diseases, such as cancer, immune-mediated diseases and neurodegenerative diseases, have lately received increasing attention[9–11,34,35]. Notably, manipulating the expression of certain miRNAs causes drastic biological changes in vivo, as demonstrated in mouse disease models[11]. In the demyelinating disorder, MS, it has been shown that lymphocyte populations derived from either blood or inflammatory lesions accompany altered expression of certain miRNAs[12]. Overabundance of various miRNAs was also confirmed in the serum or plasma samples of patients with MS[36–41]. MiRNAs carried by exosomes were shown to contribute to intercellular communications, in which exosomes would transport and transduce the miRNAs into the target cells, either close or distant from the cells of exosome origin[15]. The exosome-derived miRNAs would actually alter the gene expression and functions of the recipient cells[13,14]. For example, exosomal miRNAs derived from breast cancer cells were shown to damage the integrity of the blood–brain barrier and promote metastasis to the brain, accounting for their preferential metastasis to the brain[21].

**Fig. 5** Differentiation of Treg cells from naive CD4$^+$ T cells is inhibited by MS-exosome and *let-7i*. **a** Gating strategy for isolation of naive CD4$^+$ T cells (CD45RA$^+$CD25$^-$), memory CD4$^+$ T cells (CD45RA$^-$CD25$^-$), resting Treg cells (CD45RA$^+$CD25$^+$) and activated Treg cells (CD45RA$^-$CD25$^{high}$). **b** Each population sorted as in **a** was cultured with HC-exosome or MS-exosome under stimulation with anti-CD3 and anti-CD28 mAbs for 72 h. TGFβ (1 ng/mL) and IL-2 (50 U/mL) were added for the culture of naive CD4$^+$ T cells. The frequency of IFN-γ$^-$IL-17A$^-$Foxp3$^+$CD4$^+$ T cells was analysed for cultured naive and memory CD4$^+$ T cells. The frequencies of dead and proliferated cells were analysed for cultured resting and activated Treg cells. **c** Each population sorted as in **a** was transfected with *let-7i* or a negative control, and cultured in the same way as in **b**. Data are representative of two independent experiments. **d** Naive CD4$^+$ T cells were sorted from the peripheral blood of HC or patients with MS. Mature and primary *let-7i* were quantified by RT-qPCR. Correlation between the amount of *let-7i* in exosomes in the blood and the ratio of mature to primary *let-7i* in naive CD4$^+$ T cells was analysed. $n = 3$ in each group in **c**. An unpaired $t$ test was used in **b**, **c** and **d**, and Pearson's correlation analysis was used in **d** for statistical analysis. Error bars represent the mean ± s.d. in **b** and **d**, and the mean ± s.e.m. in **c**. s.d. standard deviation, s.e.m. standard error of the mean, n.s. not significant, A.U. arbitrary unit, rTreg resting regulatory T cells, aTreg activated regulatory T cells, RT-qPCR reverse transcription quantitative polymerase chain reaction

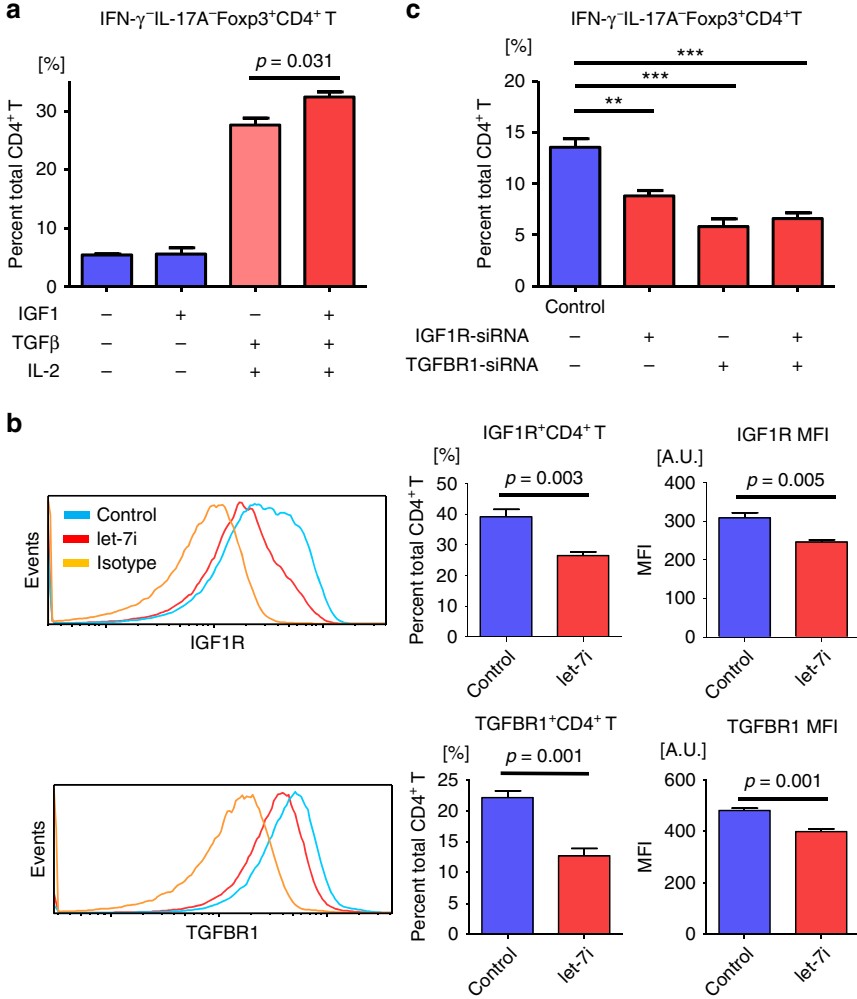

**Fig. 6** *Let-7i* transfection decreases expression of IGF1R and TGFBR1 on CD4+ T cells. **a** Induction of Treg cells in the presence of IGF1, TGFβ and IL-2. Naive CD4+ T cells were cultured in the presence of various combinations of IGF1 (10 ng/mL), TGFβ (1 ng/mL) and IL-2 (50 U/mL) under stimulation with anti-CD3 and anti-CD28 mAbs for 72 h. The frequency of IFN-γ−IL-17A−Foxp3+CD4+ Treg cells among CD4+ T cells was determined. **b** T cells were transfected with *let-7i* and then cultured under stimulation with anti-CD3 and anti-CD28 mAbs for 72 h. Representative histograms of the expression levels of IGF1R and TGFBR1 are shown. The expression levels of IGF1R and TGFBR1 were determined by measuring the frequency of each receptor-positive cells among CD4+ T cells and the MFI of each receptor on CD4+ T cells. **c** Naive CD4+ T cells were transfected with siRNAs targeting *TGFBR1* and *IGF1R* as indicated, and then differentiated towards Treg cells with TGFβ (1 ng/mL) and IL-2 (50 U/mL) in addition to stimulation with anti-CD3 and anti-CD28 mAbs for 72 h. The frequency of Treg cells was evaluated. Data are representative of two independent experiments. n = 3 **a** or 4 **b** and **c** in each group. An unpaired *t* test was used in **a** and **b**, and a one-way ANOVA with Dunnett's comparison test was used in **c** for statistical analysis. Error bars represent the mean ± s.e.m. **p < 0.01, ***p < 0.001. s.e.m. standard error of the mean, A.U. arbitrary unit, MFI mean fluorescence intensity, ANOVA analysis of variance, IGF1R insulin like growth factor 1 receptor, TGFBR1 transforming growth factor beta receptor 1

In the present study, we investigated the involvement of exosomal miRNA in MS pathogenesis. We first showed that exosomes derived from patients with MS have potentials to reduce the relative frequency of IFN-γ−IL-17A−Foxp3+CD4+ Treg cells in culture by inhibiting their differentiation from naive CD4+ T cells (Figs. 1 and 5). Subsequent analysis of the expression of exosomal miRNAs has demonstrated that the miRNA expression profiles in MS-derived exosomes are characterised by over-abundance of four miRNAs: *let-7i, miR-19b, miR-25* and *miR-92a* (Fig. 2). Notably, expression of *let-7i* in exosomes inversely correlated with the frequency of Treg cells in the culture of T cells with exosomes (Fig. 3), and we found that this miRNA is capable of reducing the frequency of the functional Treg cells (IFN-γ−IL-17A−Foxp3+CD4+ Treg) via inhibition of their differentiation from naive CD4+ T cells (Figs. 4 and 5).

It is widely recognised that a reduction of Foxp3+ Treg cell frequency is an immunological hallmark of MS[5,6]. To identify the target molecules of *let-7i*, we used an online database for target prediction and found that insulin like growth factor 1 receptor (*IGF1R*) and transforming growth factor beta receptor 1 (*TGFBR1*) are potential targets of *let-7i*. Actually, we have revealed that the expression of IGF1R and TGFBR1 on T cells was downregulated after *let-7i* transfection (Fig. 6b), whereas targeting *IGF1R* and *TGFBR1* by siRNA inhibited the differentiation of Treg cells from naive T cells in vitro (Fig. 6c). Flow cytometry analysis revealed that not only a reduction of Treg cell frequency, but also reduced expression of IGF1R and TGFBR1 on naive T cells, are characteristics of the peripheral blood of patients with MS (Fig. 8a, c). Given that MS-derived exosomes and *let-7i* could inhibit the in vitro induction of IFNγ−IL-17A−Foxp3+CD4+ Treg cells by acting on naive CD4+ T cells, the observed changes in naive T cells from the peripheral blood of patients with MS suggested that exosomal *let-7i* may also target naive T cells in vivo. Consistent with this, the Treg cell frequency in the

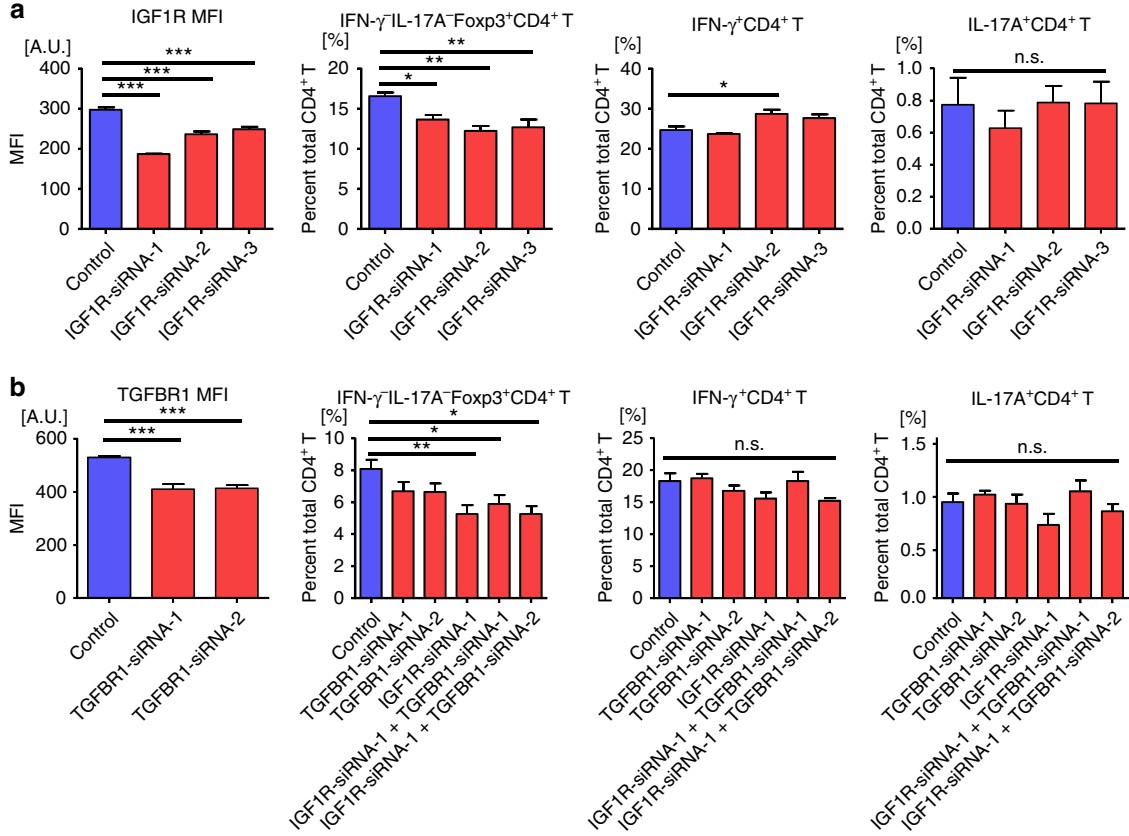

**Fig. 7** SiRNA-mediated knockdown of *IGF1R* and *TGFBR1* decreases Treg cell frequency. **a** T cells were transfected with three different siRNAs targeting *IGF1R*, and then cultured under stimulation with anti-CD3 and anti-CD28 mAbs for 72 h. The expression of IGF1R was assessed by the MFI of the receptor on CD4+ T cells. The frequencies of inflammatory and regulatory T cells among CD4+ T cells were evaluated. **b** T cells were transfected with two siRNAs targeting *TGFBR1*, with or without one targeting *IGF1R*, and cultured in the same manner as in **a**. The expression of TGFBR1 and the frequencies of inflammatory and regulatory T cells among CD4+ T cells were evaluated similarly. Data are representative of two independent experiments. *n* = 4 in each group. A one-way ANOVA with Dunnett's comparison test was used for statistical analysis. Error bars represent the mean ± s.e.m. \**p* < 0.05, \*\**p* < 0.01, \*\*\**p* < 0.001. s.e.m. standard error of the mean, A.U. arbitrary unit, MFI mean fluorescence intensity, ANOVA analysis of variance, IGF1R insulin like growth factor 1 receptor, TGFBR1 transforming growth factor beta receptor 1

peripheral blood was relatively low in persons with a higher amount of exosomal *let-7i* compared to those with a lower amount (Fig. 8d). However, we observed no significant inverse correlation between them, which might indicate the alteration of other factors involved in the homeostasis of Treg cells. Further studies with a larger sample size are needed to clarify the level of involvement of such additional factors in the homeostasis. More strikingly, the Treg cell frequency was significantly correlated with the molecules expressed by naive CD4+ T cells: *let-7i*, TGFBR1 and IGF1R (Fig. 8e, f). The overall results, including the upregulation of *let-7i* in the exosomes and downregulation of TGFBR1 and IGF1R on naive T cells, raised the possibility that alterations in the *let-7i-IGF1R/TGFBR1* axis might underlie the autoimmune pathogenesis of MS. There might be several decreased miRNAs besides the four increased miRNAs validated in Fig. 2, and possibly other differentially expressed nucleic acids, proteins and lipids in circulating exosomes in patients with MS. The relevance of these alterations to the phenotypic change in T helper cells needs to be studied further.

According to previous studies, ~30 miRNAs are differentially expressed in the sera or plasma samples of patients with MS as compared with those of HC[36–41]. The function of these cell-free circulating miRNAs has not been mentioned or has only been speculated based on pathway analysis[39–41]. However, individual miRNAs act against various cells equipped with divergent mRNA profiles, which makes the situation complicated. Therefore, it is

hard to evaluate the relevance of speculated functions of each miRNA, unless functional validation has been made. In the present work, the functions of *let-7i* against human lymphocytes have been experimentally validated and correlated with autoimmune pathogenesis. Gandhi et al. previously described upregulation of *miR-25*[40] and *miR-92a*[39] in the sera or plasma of patients with MS, although it is unclear whether these changes reflect the altered exosomal miRNA profiles. In contrast, upregulation of *let-7i* and *miR-19b* in the circulation have not been previously described regarding MS pathogenesis, indicating the need for analysing exosomal miRNAs. Among the miRNAs upregulated in the exosomes from patients with MS, *let-7i* is distinguishable from the others, in that the other three can be grouped in the same cluster or family. *MiR-19b* and *miR-92a* belong to both the *miR-17-92* and *miR-106a-363* clusters[42]. Since *miR-25* and *miR-92a* belong to the same family of miRNAs[42], we speculated that *miR-19b*, *miR-25* and *miR-92a* in exosomes may have some function distinct from that of *let-7i* in MS pathogenesis, although we have not experimentally addressed this possibility yet.

Although our results showed that homeostasis of Treg cells could be regulated by exosomal *let-7i*, it is possible that other immune cells may be an additional target of *let-7i*. In fact, a previous study demonstrated that downregulation of *let-7i* in lipopolysaccharide-stimulated dendritic cells (DC) promoted an expansion of Treg cells via upregulation of suppressor of cytokine

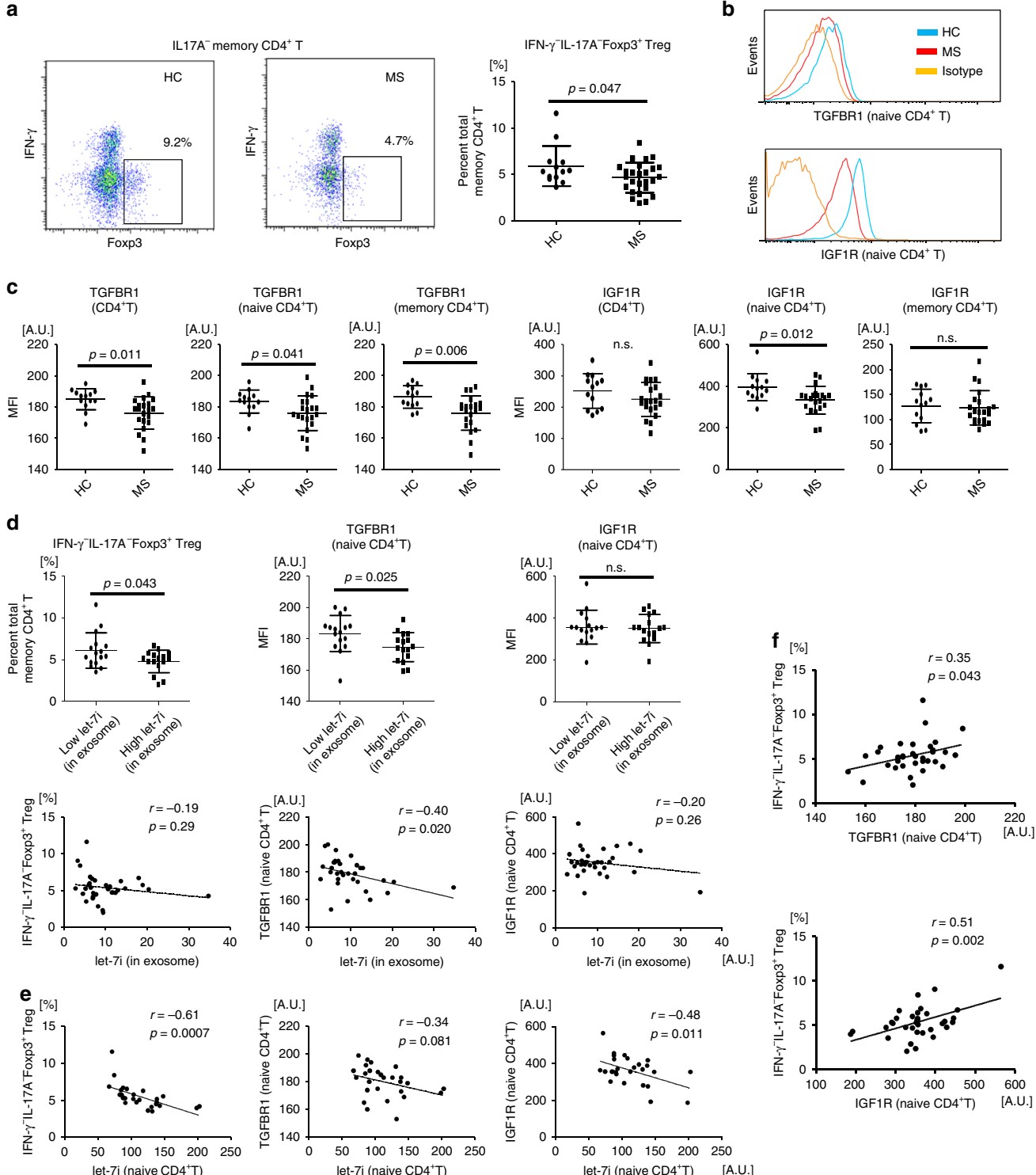

**Fig. 8** Treg cell frequency and expression of IGF1R and TGFBR1 by naive CD4+ T cells is decreased in MS. **a** The frequency of IFN-γ−IL-17A−Foxp3+ Treg cells among memory CD4+ T cells in circulation. **b** Representative histogram showing the expression levels of IGF1R and TGFBR1 on CD45RA+ naive CD4+ T cells. **c** The expression levels of IGF1R and TGFBR1 were assessed by the MFI of the receptors on circulating CD4+ T cells. **d** The frequency of IFN-γ−IL-17A−Foxp3+ Treg cells and the expression of TGFBR1 and IGF1R on naive CD4+ T cells were analysed between the groups with lower and higher amounts of *let-7i* in exosomes in the blood. The groups were determined so as to be numerically equal. Correlation analysis was also performed. **e** Correlation analysis between the amount of *let-7i* in naive CD4+ T cells and the frequency of IFN-γ−IL-17A−Foxp3+ Treg cells or the expression of TGFBR1 or IGF1R on naive CD4+ T cells in patients with MS and HC. **f** Correlation analysis between the frequency of IFN-γ−IL-17A−Foxp3+ Treg cells and the expression of TGFBR1 or IGF1R on naive CD4+ T cells in patients with MS and HC. An unpaired *t* test was used in **a**, **c** and **d**, and Pearson's correlation analysis was used in **d**, **e** and **f** for statistical analysis. Error bars represent the mean ± s.d. s.d. standard deviation, n.s. not significant, A.U. arbitrary unit, MFI mean fluorescence intensity, IGF1R insulin like growth factor 1 receptor, TGFBR1 transforming growth factor beta receptor 1

signalling 1 (SOCS1)[43], indicating that the decrease of Treg cell frequency in patients with MS may be partly explained by the effects of exosomal let-7i on DCs. The results of previous studies addressing the immunoregulatory function of let-7 family also raise the possibility that let-7i may target invariant NKT (iNKT) cells[44], IL-10-producing T cells[45] or macrophages/microglia[46]. Notably, iNKT cell frequency is reduced in the peripheral blood of patients with MS[47,48], whereas let-7 promotes differentiation of IFN-γ-producing NKT1 cells but suppresses IL-4-producing NKT2 cells by targeting a transcription factor PLZF[44].

Regarding the involvement of extracellular vesicles (EV) in MS pathogenesis, we may refer to a few reports indicating a pathogenic function of microvesicles in MS. Microvesicles, the EVs larger than exosomes, derived from platelets or endothelial cells were increased in the circulation of patients with MS[49]. Such MS patient-derived microvesicles reportedly disrupted endothelial barrier in vitro[49], and microvesicles from endothelial cells promoted the proliferation of CD4+ and CD8+ T cells[50]. Another study showed that microvesicles of myeloid origin were increased in the cerebrospinal fluid of patients with MS[51]. In mouse models of MS (experimental autoimmune encephalomyelitis), such microvesicles promoted inflammation in the central nervous system[51]. Although the precise mechanisms underlying these phenomena were not clear, a possible link with our work should be addressed in the future studies.

Expression levels of some circulating miRNAs have been shown to correlate with the disease course (RRMS or SPMS), severity or duration of MS[38–41], resulting in an assumption that they might be useful as potential biomarkers for MS. In contrast, the increase of the four exosomal miRNAs reported in this study was observed irrespective of the disease course (RRMS or SPMS) or clinical phases (relapse or remission) (Fig. 2), implying that exosomal miRNA profile may be persistently dysregulated in patients with MS. Based on these results, we may speculate that circulating miRNAs that correlate with disease status may be derived from inflammatory lesions or cells involved in the inflammatory process. On the contrary, persistently upregulated miRNAs might be caused by environmental stimulus, which is also persistent during the course of disease. It was reported that host-derived miRNAs in the microvesicles would modify gut microbiota and affect severity of experimentally induced colitis[52]. In turn, it was shown that nematode-derived vesicles in the gastrointestinal tract modified immunological defense of mice in vivo via transfer of nematode miRNAs[53]. These studies indicate that exosomes can have an important function in autoimmunity by mediating communication between host and gut microbiota. Several studies, including ours, have revealed alterations of gut microbiota in patients with MS and described its possible link with the immunopathogenesis[54–57]. Gut bacteria reduced in patients with MS may contain those engaged in the production of short chain fatty acids, which are responsible for the induction of Treg cells in the gut[54]. It is an open question whether alterations in the gut microbiota in patients with MS may somehow cause dysregulated exosomal miRNAs and the defective Treg cells.

## Methods

**Human samples.** We present demographics of the patients with MS and healthy volunteers who agreed to provide peripheral blood for this study in Supplementary Table 1. McDonald criteria[58] was used to make diagnosis of MS. The Ethics Committee of National Center of Neurology and Psychiatry approved the study protocol. Written informed consent was obtained from all patients.

**Isolation of exosomes.** Exosomes were purified from the plasma according to a previously described method, with some modifications[59]. Briefly, plasma samples were subjected to sequential centrifugation at $1200 \times g$ for 15 min and at $2000 \times g$ for 20 min, and supernatant was collected after the final centrifugation at $10\,000 \times g$ for 45 min. The resultant supernatant was mixed with a one-fifth volume of polyethylene glycol buffer (30% PEG 6000, 50 mmol/L HEPES, 1 mol/L NaCl). Exosomes were pelleted by centrifugation at $10\,000 \times g$ for 30 min and resuspended in phosphate-buffered saline. The size distribution of collected exosomes was determined using a NanoSight LM10 nanoparticle analysis system (NanoSight, NanoSight Ltd, UK).

**Transfer of PKH67-labelled exosomes.** Purified exosomes derived from HC were labelled using a PKH67 green fluorescent labelling kit (MINI67-1KT, Sigma-Aldrich, MO, USA). Exosomes were incubated with $2\,\mu M$ of PKH67 and washed using a 100-kDa filter (Amicon Ultra-0.5 mL, Millipore, Darmstadt, Germany) to remove excess dye. A control solution was prepared by the same procedure without exosomes. T cells were incubated with labelled exosomes or control solution for 24 h, followed by an examination using flow cytometer (FACS Canto II, BD Biosciences, NJ, USA) and fluorescence microscopy (BZ-X700, KEYENCE, Osaka, Japan).

**Cell preparation and flow cytometry.** PBMCs were isolated by density gradient centrifugation, using Ficoll–Paque PLUS (GE Healthcare Bioscience, ON, Canada). They were stained for cell surface antigens, followed by intracellular staining using Foxp3 / Transcription Factor Staining Buffer Set (00-5523-00, eBiosciences, CA, USA) or BD Cytofix Fixation Buffer/BD Phosflow Perm Buffer III (BD Biosciences, CA, USA) according to the manufacturer's instructions. The antibodies and isotype controls used in this study were as follows: FITC-anti-Foxp3 (PCH101, diluted 10 times (10×)), FITC-rat IgG2aκ (eBR2a, (10×)) (eBiosciences, CA, USA), PE-anti-STAT1 (1/Stat1, 10×), PE-anti-pSTAT1 (4a, 10×), PE-anti-IL-10 (JES3-19F1, 2×), PE-mouse IgG1κ (MOPC-21, 10×), PerCP-Cy5.5-anti-CD49d (9F10, 5×), PE-Cy7-anti-CD45RA (L48, 10×), APC-H7-anti-CD45RA (HI100, 10×) (BD Biosciences, CA, USA), PE-anti-CD3 (UCHT1, 20×), PE-anti-IFN-γ (B27, 200×), PE-anti-IGF1R (1H7/CD221, 10×), PE-anti-GM-CSF (BVD2-21C11, 5×), PerCP-Cy5.5-anti-CD3 (UCHT1, 10×), PerCP-Cy5.5-anti-CD4 (RPA-T4, 20×), PerCP-Cy5.5-anti-CD45RA (HI100, 20×), PE-Cy7-anti-CD25 (BC96, 20×), PE-Cy7-mouse IgG1κ (MOPC-21, 10×), APC-anti-CD8 (SK1, 10×), APC-anti-IL-17A (BL168, 10×), APC-mouse IgG1κ (MOPC-21, 10×), APC-Cy7-anti-CD8 (RPA-T8, 10×), BV421-anti-CD127 (A019D5, 10×), BV510-anti-CD8 (SK1, 10×), BV510-anti-CD45RA (HI100, 10×) (all from BioLegend, CA, USA), APC-anti-rabbit IgG (711-136-152, 3.3×) (Jackson Immuno Research Lab, PA, USA), purified anti-TGFBR1 (ab31013, 3.3×) (Abcam, Cambridge, UK). Proliferation and dead cells were analysed using CellTrace Violet Cell Proliferation Kit (C34557, Thermo Fisher Scientific, MA, USA) and Zombie Aqua Fixable Viability Kit (423101, BioLegend, CA, USA). Cells were analysed using FACS Canto II flow cytometer (BD Biosciences, CA, USA) and FlowJo software (Tree Star, OR, USA).

CD3+ T cells and naive CD4+ T cells were prepared using a pan T cell isolation kit, Naive CD4+ T Cell isolation kit, and autoMACS Pro Separator (all from Miltenyi Biotec, Germany) according to the manufacturer's instructions.

**Cell culture.** To determine the functions of exosomes and miRNAs, T cells were cultured with exosomes or after being transfected with miRNA mimics, inhibitors and siRNAs as indicated. AIM V medium (Life Technologies, CA, USA) was used for the culture. The exosomes were derived from the same plasma and the T cells were prepared from HC or patients with MS. Culture plates were coated with $2\,\mu g/mL$ of anti-CD3 mAb (OKT3) and $4\,\mu g/mL$ of anti-CD28 mAb (Beckman Coulter, CA, USA) for 1 h at 37 °C before cells were added. In the differentiation assay, naive CD4+ T cells were polarised towards Treg cells with 1 ng/mL of TGFβ and 50 U/mL of IL-2 (PeproTech, NJ, USA). An IGF1 recombinant human protein (Thermo Fisher Scientific, MA, USA) was added in some experiments as indicated. To neutralise TGFβ in the culture, $10\,\mu g/mL$ of anti-TGFβ antibody (1D11.16.8) was used. Resting and activated Treg cells, which were transfected with a mimic of let-7i or a negative control, were cultured with five times as many CD3+ T cells without transfection. To stain intracellular cytokines, cultured cells were stimulated with 50 ng/mL of phorbol-myristate-acetate (Sigma-Aldrich, MO, USA), 500 ng/mL of ionomycin (Sigma-Aldrich) and $2\,\mu M$ monensin (Sigma-Aldrich) for 4 h just before staining.

For suppression assay, CD25−CD45RA−CD4+ T cells and CD25+CD127−CD49d−CD4+ T cells were sorted after culture under stimulation with anti-CD3 and anti-CD28 mAbs for 48 h. CD45RA+CD25−CD4+ T cells were sorted as responder cells from CD3+ T cells cultured with 1 U/mL of IL-2 for 48 h. CD3−CD56− cells were sorted as APCs from PBMCs cultured without stimulation for 48 h. Responder cells $(1 \times 10^4)$ were stained with Violet Cell Proliferation Kit (C34557, Thermo Fisher Scientific, MA, USA), and cultured with CD25−CD45RA−CD4+ T cells or CD25+CD127−CD49d−CD4+ T cells $(1 \times 10^4)$ or without any additional cells (control), in the presence of APCs $(1 \times 10^5)$ under stimulation with anti-CD3 mAb for 120 h in RPMI medium (Thermo Fisher Scientific, MA, USA) containing 10% of fetal calf serum (MP Biomedicals, CA, USA). Percentage of suppression was determined based on division indices calculated with FlowJo software (Tree Star).

**Transfection of nucleic acid reagents.** MiRNA mimics, inhibitors and siRNAs were transfected using a P3 Primary Cell 4D-Nucleofector X Kit L and 4D-Nucleofector system (LONZA, Basel, Switzerland) according to the manufacturer's

protocol. Five hours after transfection, cells were seeded onto a culture plate. Materials used for transfection in this study were as follows: MISSION Human miRNA Mimics of hsa-let-7i-5p, hsa-miR-19b-3p, hsa-miR-25-3p and hsa-miR-92a-3p, and MISSION miRNA Negative Control 2 (all from Sigma-Aldrich, MO, USA); mirVana microRNA inhibitor of hsa-let-7i-5p and a negative control (Thermo Fisher Scientific, MA, USA); Silencer Select validated siRNAs of TGFBR1 and IGF1R, and Silencer Select Negative Control No. 1 siRNA (all from Thermo Fisher Scientific, MA, USA).

**RNA isolation and detection of miRNA and mRNA**. Using Plasma/Serum Circulating and Exosomal RNA Purification Kit (51000, Norgen Biotek Corp., ON, Canada), total RNA was extracted from the purified exosomes that were obtained from the same amount of the plasma. 5 pg of celmiR-39 mimic (Applied Biosystems, CA, USA) was added as an external control. RNA was subjected to complementary DNA (cDNA) synthesis using a TaqMan MicroRNA Reverse Transcription Kit (4366597, Applied Biosystems, CA, USA) and PCR was performed using TaqMan Fast Advanced Master Mix and StepOnePlus realtime PCR system (both from Thermo Fisher Scientific, MA, USA). The data of each miRNA was analysed by the delta-delta Ct method using data of celmiR-39 and normalised to the amount of applied RNA. The amount of total RNA was determined using QuantiFluor RNA system (Promega, WI, USA). For detection of cellular RNA, miRNeasy kit (217004, Qiagen, Venlo, Netherlands) was used in the extraction step. MiRNA was then analysed with the same method as described above, and the data was normalised to the expression of U6 snRNA. For analysis of mRNA or primary miRNA, extracted RNA was subjected to cDNA synthesis using PrimeScript RT Master Mix (TAKARA, Shiga, Japan) and PCR was performed using TaqMan Fast Advanced Master Mix and StepOnePlus realtime PCR system. The data was normalised to the expression of GAPDH (glyceraldehyde-3-phosphate dehydrogenase). These primers were used: TaqMan MicroRNA Assays of hsa-let-7i-5p, hsa-miR-19b-3p, hsa-miR-25-3p, hsa-miR-92a-3p and U6 snRNA; TaqMan Pri-miRNA Assay of has-let-7i; and TaqMan Gene Expression Assay of GAPDH (Applied Biosystems, CA, USA).

**Methylation analysis of *FOXP3* DNA**. Genomic DNA was extracted as follows. FACS-sorted cells were resuspended and lysed in 500 μL of lysis buffer containing 100 mM NaCl, 50 mM Tris HCl, 20 mM EDTA, 1% SDS and 0.1 mg/mL proteinase K. The lysate was incubated in a thermoshaker at 50 °C for 14 h, and then subjected to DNA extraction with 500 μL of mixture of phenol, chloroform and isoamyl alcohol (25:24:1) (Nakalai tesque, Kyoto, Japan). After centrifugation at 20,000 × g for 5 min, the aqueous (upper) portion was collected and subjected to reextraction with 500 μL of chloroform (WAKO, Osaka, Japan). After centrifugation at 20,000 × g for 10 min, the aqueous (upper) portion was mixed with 500 μL of 2-propanol (WAKO, Osaka, Japan), and centrifuged at 20,000 × g for 10 min. The precipitated pellets were rinsed with 100 μL of 70% ethanol, and centrifuged at 20,000 × g for 5 min. The resultant DNA pellets were resolved in DNase-free water. The methylation status of extracted DNA was examined (Alliance Biosystems, Osaka, Japan). Briefly, after bisulfite conversion using an EZ DNA Methylation-Lightning kit (D5030, Zymo Research, CA, USA), bisulfite-treated DNA was subjected to PCR with primers for amplification of target regions of interest. The percentage methylation was evaluated by pyrosequencing using a PyroGold reagent kit and PyroMark Q96 ID (972804, Qiagen, Venlo, Netherlands). The analysed CpG sites were located in the STAT5-responsive region of the *FOXP3* gene, and the same as the sites that were examined by the previous report[23]. The sequences of used primers are as follows: PCR primers: 5′-GTTAAGTTTGTTGTAGGA-TAGGGTAGT-3′ and 5′-AAATCTACATCTAAACCCTATTATCACA-3′; sequence primer: 5′-GTGGTGTAGATGAAGT-3′.

**Western blot**. Proteins were extracted from exosomes and cells using RIPA Lysis and Extraction Buffer (Thermo Fisher Scientific, MA, USA) containing Protease/Phosphatase Inhibitor Cocktail (Cell Signaling Technology, MA, USA). The extracted proteins were separated in XV PANTERA Gel (12.5%) (DRC, Osaka, Japan) and blotted onto polyvinylidene fluoride membranes (Merck Millipore, Darmstadt, Germany). The membranes were blocked with 5% w/v Bovine Serum Albumin (Sigma-Aldrich, MO, USA), followed by incubation overnight with primary antibodies as follows: anti-CD9 (SN4 C3-3A2, 500×, eBiosciences, CA, USA), anti-CD63 (H5C6, 500×, BioLegend, CA, USA) and anti-Cytochrome c (D18C7, 1000×, Cell Signaling Technology, MA, USA). The membranes were then incubated with secondary antibodies (HRP-linked anti-mouse IgG, A4416 or HRP-linked anti-rabbit IgG, A6154, Sigma-Aldrich, MO, USA), and digital images were obtained using ImageQuant LAS 500 (GE Healthcare, Bucks, UK) with an Immobilon Western Chemiluminescent HRP substrate (Millipore, Darmstadt, Germany).

**Microarray analysis of miRNA**. Total RNA was extracted from the purified exosomes as described above. The RNA was subjected to a comprehensive gene expression analysis using the 3D-Gene Human miRNA Oligo chip ver. 20, which targets ~2500 miRNAs (TORAY, Tokyo, Japan). The 3D-Gene miRNA labelling kit (TORAY, Tokyo, Japan) was used to label the RNA. Hybridisation signals were examined using a 3D-Gene Scanner (TORAY, Tokyo, Japan).

**Statistical analysis**. Data were analysed with FlowJo software (Tree Star) and Prism software (GraphPad Software, CA, USA). An unpaired two-tailed $t$ test was used to compare data from two groups, using Welch's correction when variances are unequal. A one-way ANOVA with Dunnett's or Bonferroni's comparison test was used to compare data from more than two groups, as appropriate. Pearson's analysis was used to evaluate correlations. Differences were considered significant when the $p$ value was < 0.05.

**Data availability**. Microarray data that support the findings of this study have been deposited in the NCNI Gene Expression Omnibus database with the primary accession code GSE86863.

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

## Acknowledgements

This study was supported by Research Project on Rare and Intractable Diseases, Health, Labour and Welfare Sciences Research, the Ministry of Health, Labour and Welfare of Japan (H26-itaku(nan)-ippan-019); and the Practical Research Project for Rare/Intractable Diseases from Japan Agency for Medical Research and Development, AMED (15ek0109097h0001, 16ek0109097h0002).

## Author contributions

K.K., H.H. and T.Y. contributed study concept and design. K.K., H.H., M.F., W.S., S.O., C.T., H.Y., T.K., R.T. and T.Y. acquired, analysed or interpreted data. K.K., H.H., M.F., W.S., S.O., T.K., R.T. and T.Y. drafted and revised the manuscript.

## Additional information

**Competing interests:** W.S. received grant support from Novartis Pharmaceuticals. T.K. received grants or research support from Bayer Holding Ltd., Takeda Pharmaceutical Co. Ltd., Chugai Pharmaceutical Co. Ltd., Novartis Pharmaceuticals, Mitsubishi Tanabe Pharma Corporation and Japan Blood Products Organization. R.T. served as a consultant for KAN Research Institute Inc., and Dainippon Sumitomo Pharma; received grants or research support from Dainippon Sumitomo Pharma, Otsuka Pharmaceutical Co., Novartis, Nihon Medi-Physics Co. Ltd., and KAN Research Institute Inc. T.Y. served on scientific advisory boards for Takeda Pharmaceutical Co. Ltd. and Chugai Pharmaceutical Co. Ltd.; received research support from Ono Pharmaceutical Co. Ltd., Chugai Pharmaceutical Co. Ltd., Biogen Idec, Novartis Pharmaceuticals, Nihon Pharmaceutical Co. Ltd., GlaxoSmithKline Co., Teva Pharmaceutical K.K., and Asahi Kasei Kuraray Medical Co. Ltd.; received speaker honoraria from Chugai Pharmaceutical Co. Ltd., Ono Pharmaceutical Co. Ltd., Takeda Pharmaceutical Co. Ltd., Biogen Idec, Dainippon Sumitomo Pharma Co. Ltd., Mitsubishi Tanabe Pharma Corporation, Yakult Bio-Science Foundation, Human Metabolome Technologies Inc., Bristol-Myers Squibb Co., and Bayer Holding Ltd. The remaining authors declare no competing financial interests.

