## [Peer Review File · Nature Communications]

Reviewers' comments:

Reviewer #1 (Remarks to the Author):

The manuscript by Kimura et al. demonstrated distinct miRNA expression profiles in circulating exosomes isolated from multiple sclerosis (MS) patients. In particular, they showed that elevated levels of exosomal miRNA, let-7i from MS patients could suppress TGFb-dependent Treg cell induction through targeting TGFBR1 and IGF1R and that TGFBR1 expression in naive CD4+ T cells was reduced in MS and significantly correlated with Treg cell frequency. While the basic ideas provided in this study are certainly of interest, nevertheless, there is a legitimate question regarding the interest level of this present study. Both TGFBR1 and IGF1R have been recently shown to be directly targeted by let-7i (the authors should include those references). Therefore, the novelty of this study relies heavily on the assumption that exosomal let-7i acquired by T cells in MS patients plays a major role in repressing TGFBR1 and IGF1R in vivo, for which there is limited data presented in this current manuscript. Many conclusions were drawn based on the correlative evidence without any direct experimental support. Therefore, without committing substantial amount of efforts to address the points made below, I am afraid that I cannot fully support the current manuscript for this journal.

Specific experimental points:

1. In Fig. 1, the authors should include a group culturing without any exosome.
2. The results in Fig. 2 showed that no significant difference in exosomal miRNA profiles between different MS subtypes. What are the frequencies of IFN γ - IL-17A- Foxp3+ CD4+ T cells in those different MS subtypes as well as in all patients with MS compared to healthy controls?
3. While the let-7i levels in exogenously added exosomes negatively correlated with the frequency of Treg cells after in vitro culture (Fig. 3), did the amounts of exosomal let-7i also negatively correlate with Treg frequencies in vivo?
4. Since there was no difference in the amounts of total exosomal RNAs isolated from HC and patients with MS (Fig. 3b), in addition to the miRNAs that were upregulated in MS exosomes, many miRNAs would be downregulated (Fig.1a,b). To this end, the ones that are downregulated might also contribute to the observed Treg phenotypes. This possibility should be at least discussed if not experimentally proven.
5. In Fig.3, the authors measured the amounts of miRNAs in exosomes. What were the levels of let-7i in T cells with or without co-culturing with MS-exo or HC-exo? In other words, how much exosomal let-7i was acquired by T cells in each group?
6. In Fig.4a left panel, the authors showed that let-7a inhibitor treatment would lead to increased IFN γ - IL-17A- Foxp3+ CD4+ T cell frequencies even in the presence of HC-exosomes. Did these results suggest that exosomes from healthy donor could also inhibit Treg cells in a let-7i-dependent manner? If so, do T cells culturing without any exosome contain more IFN γ - IL-17A- Foxp3+ CD4+ T cells (see point 1)? Moreover, the authors stated that the rate of Treg cell increase appeared to be greater in the presence of MS-exosomes than HC-exosomes, consistent with the notion that let-7i was more abundant in MS-exosome. I am not sure whether this conclusion can be made as the difference between these two groups does not seem to be statistically significant (Fig. 4b right panel).
7. Results shown in Fig. 5 and 6 suggested that let-7i negatively impact Treg cells through inhibiting TGFb-dependent Treg induction. If this is the case, one would expect that the differences in IFN γ - IL-17A- Foxp3+ CD4+ T cell frequencies between various siRNA treatment groups shown in Fig. 6 would disappear when TGFb is neutralized. To this end, even though there was no exogenously added TGFb in the culture, the authors should repeat these experiments with TGFb

blockade.

8. What are the let-7i levels in naive CD4 T cells from MS patients compared to those from WT donors? If reduced TGFBR1 and IGF1R amounts detected in T cells from MS patients were indeed due to let-7i-mediated repression, one would expect to detect a substantially higher level of let-7i in naive CD4 T cells isolated from MS patients. Nevertheless, even if this is the case, it remains possible that MS T cells could express higher levels of let-7i in a cell-intrinsic manner independent of acquiring any exosomal let-7i. Just because T cells can uptake let-7i (or other exosomal RNAs) in an in vitro culture system with excessive amounts of purified exosomes does not mean it would happen in vivo when the concentration of let-7i-containing exosomes is much lower. To this end, the authors should at least check the primary let-7i transcript levels in naive CD4 T cells to determine whether let-7i is more transcriptionally induced in T cells isolated from MS patients compared to HC.

Reviewer #2 (Remarks to the Author):

In this study Kimura et al. evaluated the miRNA content in exosomes from plasma of people with multiple sclerosis (MS) and healthy controls. They reveal clear differences and chose to investigate the functional relevance of the increased in 4 miRNA, namely let-7i, miR-19b, miR-25, and miR-92a, in exosomes from people with MS. They show that stimulation of T cells with exosomes of MS patients or their transfection with the miRNA let-7i resulted in the decrease in IFN γ - IL-17A- Foxp3+ CD4 T cells, a cell population they name facultative regulatory CD4 T cells. Reciprocally, a let-7i inhibitor strikingly increased the proportion of IFN γ - IL-17A- Foxp3+ CD4 T cells in the cultures. The authors went on to investigate the mechanisms underlying this effect and show that let-7i negatively impacts the protein expression of two targets, namely IGF1R and TGFBR1, both important for the in vitro accumulation of the IFN γ - IL-17A- Foxp3+ CD4 T cell population. Finally, the authors suggest the possible involvement of this pathway in people with MS as a decrease in circulating regulatory CD4 T cells correlated with decreased surface expression of IGF1R and TGFBR1 on CD4 T cells in patients.

The study is well designed and carefully executed. The data are mostly convincing and are certainly novel and important given their potential relevance for MS pathogenesis.

The data may lead to a new concept in the field, which may well be applicable to other autoimmune diseases. However, there are several issues that need to be clarified through addition of new data and/or text changes.

Major comments

1/ One important question relates to the heterogeneity of IFN γ - IL-17A- Foxp3+ CD4 T cell population analyzed. Indeed, the culture conditions used including anti-CD3/anti-CD28 stimulation and analysis at 48-72hrs results in previously Foxp3-negative non-regulatory CD4 T cells to express Foxp3 protein without exhibiting regulatory properties (see *Int Immunol* 2007; 19: 345; *Eur J Immunol.* 2007; 37: 129;...). Therefore, it seems impossible to decipher on which CD4 cell subset(s) the exosomes and the let-7i miRNA are acting. Gating out IFN γ + and IL-17A+ CD4 T cells, as performed in the current study, only partly addresses this issue. One way to experimentally tackle the issue is to purify the various CD4 T cell subsets, such as resting Tregs, activated Tregs, antigen-experienced CD4 T cells, naive CD4 T cells, prior to culture with exosomes, miRNA, or miRNA inhibitors. Clarifying whether MS exosomes and the let-7i miRNA prevent conversion of conventional CD4 T cells into Tregs and/or whether they blunt the in vitro proliferation - survival of bona fide Tregs is an important point to clarify.

2/ The functional properties of the IFN γ - IL-17A- Foxp3+ CD4 T cell population, named 'facultative' Treg population is unknown. The cells were not testing functionally. The methylation status of the TSDR region of the FOXP3 locus was not tested. At this stage naming them Treg cells is an over-interpretation. The term 'facultative' Treg cells is ambiguous and should be removed from the

manuscript.

3/ Another intriguing issue relates to the cellular sources of the exosomes from MS patients. Has the MS-exosome RNA profiling shed some light as to their origin?

4/ Figure 2.

On which criteria were the 4 healthy controls and the 4 MS patients involved in the miRNA profiling selected.

The remarkable stability of the exosomal miRNA profile in various subgroups of MS patients raises the question of whether this profile arises early or prior to disease onset. The authors may benefit from studying patients with clinically isolated syndromes.

5/ Figure 7.

The quantification of Treg cells among CD4 T cells based on two molecules, CD25 and CD127 is probably not stringent enough. The authors may want to adopt the strategies described by Miyara et al. *immunity* 2009 and/or Dominguez-Villar et al. *Nature Medicine* 2011 for more accurate identification of this CD4 T cell subset. It is unclear whether the MS patients studied here were treated or treatment naive, and whether they were in an active phase of disease.

The observation of decreased expression of IGF1R and TGFBR1 on naive CD4 T cells from MS patients is striking. Is there a correlation with the let-7i content of their exosomes?

How do the authors interpret the strong correlation shown in figure 7d between Treg frequency and TGFBR1 levels in naive CD4 T cells?

Reviewer #3 (Remarks to the Author):

Kimitoshi Kimura et al shows that serum exosomes from Multiple sclerosis patient were abundant in microRNA let-7i, miR-19b, miR-25 and miR-92a. Among them Let7i suppressed the Treg cells by downregulating TGFBR1 and IGF1R pathways. Authors claim to have elucidated a novel role of let-7i conveyed by exosomes in the pathogenesis of MS via blocking the IGF1R/TGFBR1 pathway.

There were several downsides in the manuscript which has led me to reject this manuscript. On the experimental levels various experiments are missing which could have support the study much better, such as lack of exosomal characterisation and extensive variations in the results among different experiments. The study could have been more convincing if these effects could have been shown in an in vivo model. In addition, the authors cannot use just use one single method (Flow cytometry) to verify everything, they need to verify their findings with other techniques as well. The results as shown do not seem convincing and effects of MS patient serum derived exosomes on Treg suppression are at best modest. Therefore, I think that *Nature Communications* will be of too high impact for this study.

Reviewer #1

A point addressed in the general comment: Both TGFBR1 and IGF1R have been recently shown to be directly targeted by let-7i (the authors should include those references).

Response: We respectfully cited the references in the revised manuscript with the following sentence (line 240):

“This is in consistence with the premise that the 3'untranslated regions (3'UTRs) of these genes are targets of let-7 family^{32,33}.”

We would point out that our present study does not simply repeat the let-7 mediated suppression of these receptors, but is unique in revealing that the pathway is involved in the function of helper T cells.

Comment 1: In Fig. 1, the authors should include a group culturing without any exosome.

Response: We repeated the same experiment with a different set of primary T cells and exosomes. Phosphate buffered saline (PBS) was used as a negative control without exosomes. We replaced Fig. 1d with the new results. There was only a subtle difference between exosomes from HC and exosome-free PBS, and exosomes from MS patients were unique in reducing the frequency of Treg cells.

Comment 2: The results in Fig. 2 showed that no significant difference in exosomal miRNA profiles between different MS subtypes. What are the frequencies of IFN γ - IL-17A- Foxp3+ CD4+ T cells in those different MS subtypes as well as in all patients with MS compared to healthy controls?

Response: In the old paper, we evaluated the frequency of CD25⁺ CD127⁻ CD4⁺ T cells, instead of IFN γ IL-17A⁻ Foxp3⁺ CD4⁺ T cells. However, to respond to the reviewer's comment seriously, we recollected and analyzed peripheral blood samples from a new cohort of MS and HC for enumeration of IFN γ IL-17A⁻ Foxp3⁺ CD4⁺ T cells (please see supplementary Table 1). Fig. 7 in the old manuscript was replaced by Fig. 8 containing new data during revising processes. Of note, IFN γ IL-17A⁻ Foxp3⁺ CD4⁺ T cells were decreased in patients with MS as described in Fig. 8a (new data). We were unable to show significant difference of the frequency of this Treg population among different groups of patients: RRMS, SPMS, in remission, in relapse, and with or without medication (shown below, new data). Considering the relative importance of other new data that need to be presented, we have not incorporated this information in the revised manuscript.

Comment 3: While the let-7i levels in exogenously added exosomes negatively correlated with the frequency of Treg cells after in vitro culture (Fig. 3), did the amounts of exosomal let-7i also negatively correlate with Treg frequencies in vivo?

Response: To seriously respond to this comment, we conducted additional experiments and confirmed that Treg cells were decreased in subjects with higher amounts of exosomal let-7i as compared to those with lower amounts (Fig. 8d (new data)). As the result would strengthen the message of our paper, we greatly appreciate the question from the reviewer. Furthermore, there was also a negative correlation between Treg cells and the amount of let-7i in naive CD4⁺ T cells (Fig. 8e (new data)). Although it is not clear at present how much of let-7i in naive CD4⁺ T cells was derived from exosomes, the data are supportive for the pathogenic role of let-7i in MS and worthy of note.

Comment 4: Since there was no difference in the amounts of total exosomal RNAs isolated from HC and patients with MS (Fig. 3b), in addition to the miRNAs that were upregulated in MS exosomes, many miRNAs would be downregulated (Fig.1a,b). To this end, the ones that are downregulated might also contribute to the observed Treg phenotypes. This possibility should be at least discussed if not experimentally proven.

Response: We agree with this consideration, and would again thank the reviewer for reminding us the important point of discussion. As there was a limitation of time for revision, a sufficient experimental investigation could not be performed. Following the reviewer's suggestion that we need to discuss on this point if not experimentally proven, we added sentences to the discussion in line 337:

“There might be several decreased miRNAs besides the four increased miRNAs validated in Fig. 2, and possibly other differentially expressed nucleic acids, proteins, and lipids in circulating exosomes in MS. The relevance of these alterations to the phenotypic change in T helper cells needs further studies.”

We picked up the four candidate miRNAs from the microarray data, based on the premise that the functional importance of miRNAs assigned with a high identification number (e.g. miR-2000,

miR-6000. Described by blue dots in Fig. 2b) is not firmly established as compared to that of miRNAs with a low identification number (e.g. miR-19, miR-25. Described by green dots in Fig. 2b). Most decreased miRNAs were those with a high identification number (Fig. 2b). Nevertheless, the possible contributions of these miRNAs are of sufficiently high interest and should be studied in the future.

Comment 5: In Fig.3, the authors measured the amounts of miRNAs in exosomes. What were the levels of let-7i in T cells with or without co-culturing with MS-exo or HC-exo? In other words, how much exosomal let-7i was acquired by T cells in each group?

Response: We again thank the reviewer for very valuable comments. As this reviewer described in comment 8, the ratio of mature let-7i to primary let-7i is thought to reflect the uptake of exogenous exosomal let-7i by T cells. To address the comment 5, we conducted an additional experiment and showed that the mature to primary let-7i ratio was increased in T cells cultured with MS-exosome as compared to those with HC-exosome (Fig. 3c (new data)). This result could be indirect but supportive evidence for that exosomal let-7i was actually taken up by T cells after culture with MS-exosome.

Comment 6: In Fig.4b left panel, the authors showed that let-7i inhibitor treatment would lead to increased IFN γ - IL-17A- Foxp3⁺ CD4⁺ T cell frequencies even in the presence of HC-exosomes. Did these results suggest that exosomes from healthy donor could also inhibit Treg cells in a let-7i-dependent manner? If so, do T cells culturing without any exosome contain more IFN γ - IL-17A- Foxp3⁺ CD4⁺ T cells (see point 1)? Moreover, the authors stated that the rate of Treg cell increase appeared to be greater in the presence of MS-exosomes than HC-exosomes, consistent with the notion that let-7i was more abundant in MS-exosome. I am not sure whether this conclusion can be made as the difference between these two groups does not seem to be statistically significant (Fig. 4b right panel).

Response: As we described in the response to comment 1, the IFN γ IL-17A⁻ Foxp3⁺ Treg frequency was similar between PBS control (no exosomes) and HC-exosome (Fig. 1d (new data)). This negative result is important, considering that exosomes include other miRNAs, nucleic acids, proteins and lipids, which might affect the phenotype of T cells. In addition, we repeated the same experiment as Fig.4b. The effect of let-7i inhibitor was less pronounced, compared with that shown in the previous manuscript, which may be explained by the use of different set of primary T cells and exosomes. However, the rate of increase in Treg frequency was significantly greater in the presence of MS-exosome than HC-exosome, which is thought to reflect higher amount of let-7i in MS-exosome (Fig. 4b right panel (new data)). The difference of Treg frequency between HC-exosome and MS-exosome was not observed after treatment of *let-7i* inhibitor (Fig. 4b left panel (new data)), suggesting that let-7i is critically involved in the decrease of Treg cells by MS-exosome as compared to HC-exosome. Fig.4b was replaced with this new result.

Comment 7: Results shown in Fig. 5 and 6 suggested that let-7i negatively impact Treg cells

through inhibiting TGFb-dependent Treg induction. If this is the case, one would expect that the differences in IFN γ - IL-17A- Foxp3⁺ CD4⁺ T cell frequencies between various siRNA treatment groups shown in Fig. 6 would disappear when TGFb is neutralized. To this end, even though there was no exogenously added TGFb in the culture, the authors should repeat these experiments with TGFb blockade.

Response: We performed the exact experiment with a TGF β neutralizing antibody following the reviewer's valuable comment. CD3⁺ T cells were transfected with five different siRNAs targeting TGFBR1 or IGF1R, or a negative control siRNA, and then cultured under stimulation with anti-CD3 and anti-CD28 mAbs for 72 h (in the same manner as in Fig. 7 (Fig. 6 in the old manuscript)). Anti-TGF β neutralizing antibody (10 μ g/ml) or control mouse IgG was added in the culture. In the absence of TGF β antibody, IFN γ IL-17A⁻ Foxp3⁺ Treg cells were decreased by all the siRNAs as compared to negative control siRNA (Supplementary Fig. 6a (new data)). In contrast, the differences in IFN γ IL-17A⁻ Foxp3⁺ CD4⁺ Treg cell frequencies disappeared by TGF β neutralization (Supplementary Fig. 6b (new data)), as the reviewer expected. We respectfully added this result in the manuscript (Supplementary Fig. 6 (new data)).

Comment 8: What are the let-7i levels in naive CD4 T cells from MS patients compared to those from WT donors? If reduced TGFBR1 and IGF1R amounts detected in T cells from MS patients were indeed due to let-7i-mediated repression, one would expect to detect a substantially higher level of let-7i in naive CD4 T cells isolated from MS patients. Nevertheless, even if this is the case, it remains possible that MS T cells could express higher levels of let-7i in a cell-intrinsic manner independent of acquiring any exosomal let-7i. Just because T cells can uptake let-7i (or other exosomal RNAs) in an in vitro culture system with excessive amounts of purified exosomes does not mean it would happen in vivo when the concentration of let-7i-containing exosomes is much lower. To this end, the authors should at least check the primary let-7i transcript levels in naive CD4 T cells to determine whether let-7i is more transcriptionally induced in T cells isolated from MS patients compared to HC.

Response: We greatly appreciate this comment, directing us to evaluate primary and mature let-7i transcript levels. We conducted the analysis and showed a trend for increased mature let-7i in naive CD4⁺ T cells isolated from patients with MS. In contrast, the amount of primary let-7i was similar between HC and MS (Fig. 5d (new data)). The ratio of mature let-7i to primary let-7i was significantly higher in naive CD4⁺ T cells derived from patients with MS (Fig. 5d (new data)). As the reviewer has kindly suggested, this result can be interpreted that an increased ratio of mature let-7i/primary let-7i in naive CD4⁺ T cells from MS possibly reflects an acquisition of exogenous let-7i. Further analysis revealed a positive correlation between this ratio and the amount of let-7i in exosomes in the circulation (Fig. 5d (new data)). These results are consistent with our postulate that exosomal let-7i is taken up by naive CD4⁺ T cells not only *in vitro* but also *in vivo*. Accordingly, we have added minor revisions to the appropriate places, where uptake of let-7i needs to be discussed.

Reviewer #2

Comment 1: One important question relates to the heterogeneity of IFN γ - IL-17A- Foxp3⁺ CD4 T cell population analyzed. Indeed, the culture conditions used including anti-CD3/anti-CD28 stimulation and analysis at 48-72hrs results in previously Foxp3-negative non-regulatory CD4 T cells to express Foxp3 protein without exhibiting regulatory properties (see Int Immunol 2007; 19: 345; Eur J Immunol. 2007; 37: 129;...). Therefore, it seems impossible to decipher on which CD4 cell subset(s) the exosomes and the let-7i miRNA are acting. Gating out IFN γ ⁺ and IL-17A⁺ CD4 T cells, as performed in the current study, only partly addresses this issue. One way to experimentally tackle the issue is to purify the various CD4 T cell subsets, such as resting Tregs, activated Tregs, antigen-experienced CD4 T cells, naive CD4 T cells, prior to culture with exosomes, miRNA, or miRNA inhibitors. Clarifying whether MS exosomes and the let-7i miRNA prevent conversion of conventional CD4 T cells into Tregs and/or whether they blunt the in vitro proliferation - survival of bona fide Tregs is an important point to clarify.

Response: We agree with the reviewer in that it is critical to determine which CD4⁺ T cell subset is affected by exosomes and let-7i. To address this critical issue, we contributed our sincere efforts in the last four months to performing the experiments suggested by the reviewer. Naive and memory CD4⁺ T cells, resting Treg, and activated Treg cells were defined as CD45RA⁺ CD25⁻, CD45RA⁻ CD25⁻, CD45RA⁺ CD25⁺ and CD45RA⁻ CD25^{high} CD4⁺ T cells, respectively, according to a previous paper (Miyara, M., et al. Immunity 30, 899-911 (2009)) (Fig. 5a (new data)). Each isolated population was cultured with HC-exosomes or MS-exosomes under stimulation with anti-CD3 and anti-CD28 mAbs for 72 h. MS-exosome inhibited the differentiation of IFN γ IL-17A⁻ Foxp3⁺ CD4⁺ T cells from naive CD4⁺ T cells as compared to HC-exosome (Fig. 5b (new data)). In contrast, there was no difference in survival and proliferation of resting Treg and activated Treg cells (Fig. 5b (new data)). Although there was a trend for decrease of IFN γ IL-17A⁻ Foxp3⁺ CD4⁺ T cells among cultured memory CD4⁺ T cells by MS-exosome, it was not statistically significant (Fig. 5b (new data)). As is consistent with this finding, *let-7i* inhibited the induction of IFN γ IL-17A⁻ Foxp3⁺ CD4⁺ T cells from naive CD4⁺ T cells, and there was no significant change in IFN γ IL-17A⁻ Foxp3⁺ CD4⁺ T cells among cultured memory CD4⁺ T cells nor in survival and proliferation of resting Treg and activated Treg cells (Fig. 5c (new data)). These results indicated the differentiation of Treg cells from naive CD4⁺ T cells is the main target of MS-exosome and let-7i. This is in line with our result that TGFBR1 and IGF1R, which are downstream targets of let-7i, are involved in differentiation of Treg cells (Fig. 6a, c). The advice from the reviewer actually helped us improve the quality of the work.

Comment 2: The functional properties of the IFN γ - IL-17A- Foxp3⁺ CD4 T cell population, named ‘facultative’ Treg population is unknown. The cells were not testing functionally. The methylation status of the TSDR region of the FOXP3 locus was not tested. At this stage naming them Treg cells is an over-interpretation. The term ‘facultative’ Treg cells is ambiguous and should be removed from the manuscript.

Response:

We appreciate this valuable comment. To convince the reviewer and other readers that the IFN γ ⁻ IL-17A⁻ Foxp3⁺ CD4⁺ T cell population possesses regulatory functions, we approached this issue by conducting additional experiments, including assessment of methylation status of the TSDR region of the FOXP3 locus. Although it is ideal to assess the regulatory function of IFN γ ⁻ IL-17A⁻ Foxp3⁺ CD4⁺ T cells directly, these cells cannot be obtained intact. Instead, CD25⁺ CD127⁻ CD49d⁻ CD4⁺ T cells were isolated after culture under stimulation with anti-CD3 and anti-CD28 mAbs for 48 h. As shown in Supplementary Fig. 1c (new data), this population is enriched in IFN γ ⁻ IL-17A⁻ Foxp3⁺ CD4⁺ T cells. For suppression assay, CD25⁻ CD45RA⁻ CD4⁺ T cells were used for a negative control because this population included smaller numbers of IFN γ ⁻ IL-17A⁻ Foxp3⁺ CD4⁺ T cells (Supplementary Fig. 1c (new data)). Responder cells (CD45RA⁺ CD25⁻ CD4⁺ T cells) were cultured with CD25⁺ CD127⁻ CD49d⁻ CD4⁺ T cells or CD25⁻ CD45RA⁻ CD4⁺ T cells, or without any additional cells. As a result, we were able to show that CD25⁺ CD127⁻ CD49d⁻ CD4⁺ T cells possess regulatory function, supporting the regulatory nature of IFN γ ⁻ IL-17A⁻ Foxp3⁺ CD4⁺ T cells (Supplementary Fig. 1d (new data)). To provide further confirmation, we analyzed DNA methylation status of TSDR of Foxp3 gene. The DNA region was moderately demethylated in IFN γ ⁻ IL-17A⁻ Foxp3⁺ CD4⁺ T cells, and clearly different from that of Foxp3⁺ CD4⁺ T cells secreting IFN γ or IL-17A, which was methylated to the similar extent as Foxp3⁻ non-Treg cells (Supplementary Fig. 1a, b (new data)). These data indicated that IFN γ ⁻ IL-17A⁻ population is at least modestly functional Treg cells among all Foxp3⁺ CD4⁺ T cells and the remaining IFN γ ⁺ or IL-17A⁺ cells are not. The term ‘facultative’ was removed from the manuscript.

Comment 3: Another intriguing issue relates to the cellular sources of the exosomes from MS patients. Has the MS-exosome RNA profiling shed some light as to their origin?

Response: We are also curious about the cellular sources of the exosomes including high amount of let-7i. The sorting machinery that is involved in packaging of miRNAs in exosomes are fine-tuned, and specific miRNA profile is observed in exosomes, which is different from the miRNA profile in parent cells (Villarroya-Beltri C., et al. Nat Commun 4, 2980 (2013), Okoye IS., et al. Immunity 41, 89-103 (2014), Zhang J., et al. Genomics Proteomics Bioinformatics 13, 17-24 (2015)). Therefore, it is difficult to deduce the sources of exosomes based on their miRNA profile. At the present moment, we would have to say it is almost impossible to detect the sources accurately.

Comment 4: Figure 2. On which criteria were the 4 healthy controls and the 4 MS patients involved in the miRNA profiling selected. The remarkable stability of the exosomal miRNA profile in various subgroups of MS patients raises the question of whether this profile arises early or prior to disease onset. The authors may benefit from studying patients with clinically isolated syndromes.

Response: Three out of four MS patients were in relapse at the acquisition of the sample, and healthy subjects were sex- and age-matched ones. Therefore, it was not anticipated, at first, that the four miRNAs were upregulated regardless of disease stage and relapse state. There remains a

possibility that whole miRNA profile is different depending on disease stage and relapse state. Although there was a limitation of time for revision and a sufficient number of patients with clinically isolated syndrome (CIS) could not be recruited, there was no significant difference in the amount of exosomal let-7i between patients with MS and CIS (Figure described below (new data)). Exosomal let-7i was significantly increased in patients with CIS as compared to HC. This result indicated that the alteration of exosomal miRNA arises at least at the very early stage of disease. However, as the data will be felt to be too preliminary by some readers, we would not like to add it to the revised manuscript. Kind understanding is very much appreciated.

Comment 5-1: Figure 7. The quantification of Treg cells among CD4 T cells based on two molecules, CD25 and CD127 is probably not stringent enough. The authors may want to adopt the strategies described by Miyara et al. *immunity* 2009 and/or Dominguez-Villar et al. *Nature Medicine* 2011 for more accurate identification of this CD4 T cell subset. It is unclear whether the MS patients studied here were treated or treatment naive, and whether they were in an active phase of disease.

Response: As a similar concern was raised by reviewer 1, we recollected and analyzed new blood samples for evaluation of more accurately defined Treg population. Dominguez-Villar et al. described that IFN γ ⁺ population among Foxp3⁺ CD4⁺ have reduced suppressive activity. Our definition of Treg cells *in vitro* as “IFN γ ⁻ IL-17A⁻ Foxp3⁺ CD4⁺ T cells” is similar to what Dominguez-Villar et al. reported. Therefore, we analyzed newly collected blood samples for IFN γ ⁻ IL-17A⁻ Foxp3⁺ CD4⁺ T cells. We have then confirmed that IFN γ ⁻ IL-17A⁻ Foxp3⁺ population was significantly decreased in patients with MS as compared to HC (Fig. 8a (new data)). We also evaluated the frequency of “Treg cells” defined by other definition: “resting and activated Treg cells (CD45RA⁺ Foxp3⁺ and CD45RA⁻ Foxp3^{high} CD4⁺ T cells, respectively)” described by Miyara et al. (Figure described below, upper (new data)) and “CD25⁺ CD127⁻ CD4⁺ T cells” which we adopted in our old manuscript. There was a positive correlation between various Treg definitions (Figure described below, lower (new data)). Notably, “IFN γ ⁻ IL-17A⁻ Foxp3⁺ CD4⁺ T cells” had a strong correlation with “resting and activated Treg cells (definition by Miyara et al.)”, supporting the utility of our definition of Treg cells as “IFN γ ⁻ IL-17A⁻ Foxp3⁺ CD4⁺ T cells”. We appreciate this

valuable comment, and have respectfully replaced Fig. 7 in the old manuscript with Fig. 8 in the revision based on the new definition of Treg cells as “IFN γ ⁻ IL-17A⁻ Foxp3⁺ CD4⁺ T cells”.

The Treg cell frequency was analyzed between various disease states (Figure described below (new data)). Although there was a trend for decrease of Treg cells in subjects in relapse, it was not significant. The left figure is about the newly recruited samples, and the right figure is about the subjects analyzed in our old manuscript. Considering the relative importance of other new data that need to be presented, we have not incorporated this information in the revised manuscript.

Comment 5-2: The observation of decreased expression of IGF1R and TGFBR1 on naive CD4 T cells from MS patients is striking. Is there a correlation with the let-7i content of their exosomes?

Response: To respond to this comment, we conducted additional experiments. As shown in Fig. 8d (new data), the expression of TGFBR1 on naive CD4⁺ T cells was decreased in subjects with higher amount of exosomal let-7i as compared to those with lower amount. In contrast, there was no significant difference in the expression of IGF1R between let-7i high and low groups. In addition,

there was also a significant negative correlation or a trend for inverse correlation between the amount of let-7i in naive CD4⁺ T cells and the expression of IGF1R or TGFBR1 on naive CD4⁺ T cells (Fig. 8e (new data)). Furthermore, we examined the ratio of mature let-7i to primary let-7i in naive CD4⁺ T cells. MiRNA genes are transcribed into primary miRNAs, which are then processed into mature miRNAs during several steps in the nucleus and the cytoplasm. Therefore, the mature let-7i/primary let-7i ratio in the cells could be higher, if they actually uptake exogenous mature let-7i. As a result, we showed the mature let-7i/primary let-7i ratio was increased in naive CD4⁺ T cells in MS, and the ratio was significantly correlated with the amount of exosomal let-7i (Fig. 5d (new data)). These results support the postulate that exosomal let-7i affects naive CD4⁺ T cells, decreases the expression of TGFBR1 and possibly IGF1R, and finally decreases the frequency of Treg cells by inhibiting their differentiation.

Comment 5-3: How do the authors interpret the strong correlation shown in figure 7d between Treg frequency and TGFBR1 levels in naive CD4 T cells?

Response: We are also very curious about the meaning of the strong correlation between Treg frequency and TGFBR1 expression on naive CD4⁺ T cells. It may be partly explained by a major contribution of TGFβ in differentiation of Treg cells from naive CD4⁺ T cells, as described in Fig. 6a, c. In the analysis using a new definition of Treg cells (IFNγ⁻ IL-17A⁻ Foxp3⁺ CD4⁺ T cells), Treg frequency positively correlated not only with TGFBR1 but also with IGF1R expression on naive CD4⁺ T cells (Fig. 8f (new data)). This change could be attributed to adoption of a new definition of Treg cells, and again this is consistent with their significant roles in differentiation of Treg cells (Fig. 6a, c).

Reviewer #3

Comment 1: On the experimental levels various experiments are missing which could have support the study much better, such as lack of exosomal characterisation and extensive variations in the results among different experiments.

Response: We are very sorry that the manuscript submitted 6 months ago was felt to be preliminary by the reviewer. However, we have vigorously conducted additional experiments and incorporated the new results in the revised manuscript. We described the additional information based on more solid data, and we have obtained an opportunity to describe all the details in this letter about how we have responded to the criticisms from the reviewers. We hope that you will reevaluate the work based on all of our efforts.

Actually, we adopted well-recognized methods for evaluation of collected particles from the plasma. Nano tracking analysis is a most common way to analyze diameters of particles such as exosomes efficiently, and western blotting is also an efficient tool to determine characteristics of extracellular vesicles (EVs) based on the expression of several proteins which are specific for exosomes or other classes of EVs (Fig. 1a, b).

Although there are not always statistically striking results throughout the study, the results are always consistent. To make the study more convincing, we performed several new important experiments. In Fig. 5d (new data), we evaluated the amount of primary let-7i and mature let-7i in naive CD4⁺ T cells in the peripheral blood. MiRNA genes are transcribed into primary miRNAs, which are then processed into mature miRNAs during several steps in the nucleus and the cytoplasm. Therefore, the mature let-7i/primary let-7i ratio in the cells could be higher, if they actually uptake exogenous mature let-7i. As a result, we showed the mature let-7i/primary let-7i ratio was increased in naive CD4⁺ T cells in MS (Fig. 5d (new data)). This implicated that exogenous mature let-7i was probably taken up by circulating T cells, irrespective of cell-intrinsic primary let-7i. Furthermore, the ratio positively correlated with the amount of mature let-7i in exosomes in the blood. This suggested that the source of let-7i that was taken up by T cells might be circulating exosomes, as is consistent with data in vitro (Fig. 3c (new data)).

In the old manuscript, we showed that both MS-exosomes and let-7i have potentials to decrease the frequency of Treg cells (IFN γ IL-17A⁻ Foxp3⁺ CD4⁺ T cells) among cultured CD3⁺ T cells. To identify which T cell subset is responsible for this alteration, we sorted naive and memory CD4⁺ T cells (CD45RA⁺ CD25⁻ and CD45RA⁻ CD25⁻ CD4⁺ T cells, respectively) and resting and activated Treg cells (CD45RA⁺ CD25⁺ and CD45RA⁻ CD25^{high} CD4⁺ T cells, respectively) according to a previous paper (Miyara, M., et al. *Immunity* 30, 899-911 (2009)) (Fig. 5a (new data)). Each isolated population was cultured with HC-exosomes or MS-exosomes under stimulation with anti-CD3 and anti-CD28 mAbs for 72 h. MS-exosome specifically inhibited the differentiation of IFN γ IL-17A⁻ Foxp3⁺ CD4⁺ T cells from naive CD4⁺ T cells as compared to HC-exosome (Fig. 5b (new data)). In contrast, there was no difference in survival and proliferation of resting and activated Treg cells (Fig. 5b (new data)). Although there was a trend for decrease of IFN γ IL-17A⁻ Foxp3⁺ CD4⁺ T cells among cultured memory CD4⁺ T cells by MS-exosome, it was not statistically significant (Fig. 5b (new data)). As is consistent with this finding, let-7i inhibited the induction of

IFN γ IL-17A⁻ Foxp3⁺ CD4⁺ T cells from naive CD4⁺ T cells, and there was no significant change in IFN γ IL-17A⁻ Foxp3⁺ CD4⁺ T cells among cultured memory CD4⁺ T cells nor in survival and proliferation of resting Treg and activated Treg cells (Fig. 5c (new data)). These results indicated the differentiation of Treg cells from naive CD4⁺ T cells is the main target of MS-exosome and let-7i. This is in line with another result that TGFBR1 and IGF1R, which are downstream targets of let-7i, are involved in differentiation of Treg cells (Fig. 6a, c).

The frequency of Treg cells were positively correlated with the expression of TGFBR1 and IGF1R on naive CD4⁺ T cells in the peripheral blood (Fig. 8f (new data)). In addition, Treg cells were shown to be decreased in subjects with a higher amount of exosomal let-7i as compared to subjects with a lower amount (Fig. 8d (new data)). Furthermore, the frequency of Treg cells had a strong negative correlation with the amount of let-7i in naive CD4⁺ T cells in the blood (Fig. 8e (new data)). This suggested that exosomal let-7i altered the homeostasis of Treg cells not only *in vitro* but also *in vivo*.

These newly added results further support the notion that exosomal let-7i is taken up by naive CD4⁺ T cells, decreases the expression of TGFBR1 and IGF1R, and finally decreases Treg cell frequency by inhibiting the differentiation in patients with MS.

Comment 2: The study could have been more convincing if these effects could have been shown in an *in vivo* model.

Response: We faithfully acknowledge that our study has a limitation in availability of *in vivo* procedures such as genetic modification and a treatment trial. However, experiments with samples from actual patients can directly reflect the disease mechanism. Therefore, one of several advantages of our study is that it handles human samples. There are of course several splendid studies that report experiments using only human samples (Rao, DA., et al. Nature 542, 110-114 (2017); Dominguez-Villar, M., et al. Nature medicine 17, 673-675 (2011); Hartmann, FJ., et al. Nature Communications 5, 5056 (2014)). We think that the *ex vivo* data described in Figure 8 together with a plenty amount of *in vitro* culture experiments, including newly added ones in the revision, are tolerable for sophisticated human studies.

Comment 3: In addition, the authors cannot use just use one single method (Flow cytometry) to verify everything, they need to verify their findings with other techniques as well.

Response: In our current study, we focused on the phenotype of T cells correlated with let-7i pathway. As the inflammatory and regulatory T cells are defined based on the expression of several molecules at the same time, the change in phenotype of T cells cannot be detected without single cell analysis by flow cytometry. There are several valuable studies that report experiments using mainly flow cytometry (Rao, DA., et al. Nature 542, 110-114 (2017); Dominguez-Villar, M., et al. Nature medicine 17, 673-675 (2011); Hartmann, FJ., et al. Nature Communications 5, 5056 (2014), the same papers as human sample studies described above). When such precise single cell analysis is not necessary, we also used various procedures, including RT-qPCR of miRNA and mRNA, microarray analysis, and western blotting. We also performed a new kind of experiments to show

the phenotype of IFN γ ⁻ IL-17A⁻ Foxp3⁺ CD4⁺ T cells in the revised manuscript. We analyzed DNA methylation status of TSDR (Treg-specific demethylated region) of Foxp3 gene. CpG sites in the region are known to be demethylated in functional Foxp3⁺ Treg cells (Miyara, M., et al. *Immunity* 30, 899-911 (2009)). The region examined was moderately demethylated in IFN γ ⁻ IL-17A⁻ Foxp3⁺ CD4⁺ T cells, and clearly different from that of Foxp3⁺ CD4⁺ T cells secreting IFN γ or IL-17A, which was methylated to the similar extent as Foxp3⁻ non-Treg cells (Supplementary Fig. 1a, b (new data)). These data indicated that IFN γ ⁻ IL-17A⁻ population includes a functional subset among all Foxp3⁺ CD4⁺ T cells and the remaining IFN γ ⁺ or IL-17A⁺ cells do not.

Reviewers' comments:

Reviewer #1 (Remarks to the Author):

The authors have largely responded to the concerns raised in my previous comments. I just have two relatively minor comments that need to be addressed before giving my full endorsement.

1. In response to my previous point #3, the authors now have included new data in Fig. 8d. However, it is uncertain to me as to how the authors defined the groups with lower and higher amounts of let-7i in exosomes in blood. It should be done in the same way as it was shown in Fig. 3a or Fig. 8e using Pearson's correlation analysis.

2. In many figures (ex. Fig. 1d), the scale of Y-axis did not start at 0. This is inappropriate as it would mislead the readers into thinking the differences are bigger than they actually are.

Reviewer #2 (Remarks to the Author):

I would like to congratulate the authors for taking to heart the comments and suggestions of this reviewer. They have performed elaborate experiments that answered all my queries satisfactorily. I have no further concern.

The study is now an important addition to the field. I therefore strongly recommend publication.

Reviewer #1

Comment 1: In response to my previous point #3, the authors now have included new data in Fig. 8d. However, it is uncertain to me as to how did the authors defined the groups with lower and higher amounts of let-7i in exosomes in blood. It should be done in the same way as it was shown in Fig. 3a or Fig. 8e using Pearson's correlation analysis.

Response: The groups were determined so as to be numerically equal. However, to seriously respond to this comment, we also performed correlation analysis as in Fig. 8e. Significant negative correlation was detected between the expression of TGFBR1 and the amount of exosomal let-7i, although negative correlation for IFN γ IL-17A $^{-}$ Foxp3 $^{+}$ CD4 $^{+}$ T cells was not significant. We respectfully inserted these results both in Fig. 8 and in the manuscript (result section).

Comment 2: In many figures (ex. Fig. 1d), the scale of Y-axis did not start at 0. This is inappropriate as it would mislead the readers into thinking the differences are bigger than they actually are.

Response: We appreciate the valuable comment of the reviewer. We respectfully changed the scale of Y-axis in Fig. 1d, Fig. 4b, Fig. 5b, Fig. 6b, and Fig. 7a, b, to make the results easier to understand. On the other hand, we feel it would be better not to change Fig. 3a and Fig. 8 for the general readers to easily recognize the correlation and the difference. We suspect that the absolute value of MFI (mean fluorescence intensity) (Fig. 8) is less important compared to that of other parameters such as population frequencies.

REVIEWERS' COMMENTS:

Reviewer #1 (Remarks to the Author):

As stated in the title, the central message of this manuscript is that circulating exosomes with higher amount of let-7i from multiple sclerosis patients promote pathogenesis by reducing Treg cells. However, this important point is considerably weakened by the fact that the new Pearson's correlation analysis included in Fig.8d has shown that the amounts of exosomal let-7i did not negatively correlate with Treg frequencies in patients with MS. Beside adding one sentence describing their negative results (line 280-281), I feel that the authors should at the minimum further address/discuss this issue and change their original statement in the discussion (line 333-334).

Reviewer #1

Comment 1: As stated in the title, the central message of this manuscript is that circulating exosomes with higher amount of *let-7i* from multiple sclerosis patients promote pathogenesis by reducing Treg cells. However, this important point is considerably weakened by the fact that the new Pearson's correlation analysis included in Fig.8d has shown that the amounts of exosomal *let-7i* did not negatively correlate with Treg frequencies in patients with MS. Beside adding one sentence describing their negative results (line 280-281), I feel that the authors should at the minimum further address/discuss this issue and change their original statement in the discussion (line 333-334).

Response: To respectfully address the comment by reviewer #1, we added further discussion:

“This discrepancy might indicate the involvement of factors other than exosomal *let-7i* in the homeostasis of Treg cells, although it is possible that validation with a larger sample size might blur the discrepancy.” in line 284.

“Consistent with this, the Treg cell frequency in the peripheral blood was relatively low in persons with a higher amount of exosomal *let-7i* compared to those with a lower amount (Fig. 8d). However, we observed no significant inverse correlation between them, which might indicate the alteration of other factors involved in the homeostasis of Treg cells. Further studies with a larger sample size are needed to clarify the level of involvement of such additional factors in the homeostasis.” in line 343.

Interindividual genetic diversity, for example, might mediate alteration of Treg cell frequency. In human studies, it is difficult to equalize complex background, other than exosomal *let-7i* in this case. The sample size might be relatively small. These factors could explain the non-significant correlation here. Nevertheless, the Treg frequency was different between persons with higher and lower amount of exosomal *let-7i*. Further study may reveal other possible contributors to the alteration of Treg cells.